# CROSS-MODAL MITIGATION OF SPURIOUS CORRELATION FOR PROMPT-TUNING IN VLMS WITH CAUSALLY MOTIVATED LOGIC ALIGNMENT

## ABSTRACT

Recent studies have shown that pre-trained vision-language models can effectively adapt to diverse downstream tasks through parameter-efficient prompt tuning. Unfortunately, the tuned models can exploit spurious correlations during prediction, resulting in a failure to generalize to out-of-distribution test data, especially when the tuning dataset exhibits bias. How to achieve cross-modal mitigation of spurious correlations during prompt tuning of vision-language models remains an open question. In this paper, the challenging problem is tackled by leveraging the stable relationship between necessary and sufficient causal features and the corresponding label. On the one hand, we constrain the learning process of prompt by reinforcing the necessary and sufficient connection between the textual labels and textual features. On the other hand, the probability of necessity and sufficiency between the textual features and the filtered visual features is measured and maximized to enhance cross-modal feature alignment. By iteratively optimizing these two objectives, we can achieve cross-modal mitigation of spurious correlations because the logic equivalence between textual labels and visual features is bolstered. The theoretical analysis on generalization error indicates that our method can achieve a tighter generalization error bound than existing approaches. We evaluate the proposed method on several commonly adopted out-of-distribution datasets, and the empirical results demonstrate the superiority of our method over the state-of-the-art competitors.

## 1 INTRODUCTION

Vision-language models (VLMs), which integrate visual and textual data processing for complex real-world tasks (Zhou et al., 2020; Radford et al., 2021; Zhao et al., 2024; Zhang et al., 2024c), have become a cornerstone of multi-modal learning. Recent advancements have demonstrated the powerful zero-shot generalization capabilities of pre-trained vision-language models (VLMs), enabling them highly adaptable to a wide range of downstream tasks, especially image classification (Radford et al., 2021). To harness the flexible adaptability of pre-trained VLMs, prompt tuning emerges as a parameter-efficient tuning technique and has achieved significant success (Zhou et al., 2022b;a; Chen et al., 2023). Rather than fine-tuning all model parameters, prompt tuning focuses on modifying the text prompts while keeping the model's pre-trained parameters largely intact. Optimizing the learnable prompts can enhance the alignment between textual and visual representations, thereby improving the performance of vision-language models.

It has been found that modern machine learning and data-driven models can easily rely on spurious correlations to make prediction (Geirhos et al., 2020; Ye et al., 2024). Referring to statistical associations between variables, spurious correlations arise from statistical bias and confounding factors rather than representing a true causal relationship. Consequently, spurious correlations are unstable and can vary across different data distributions. Thus, the performance of models utilizing spurious correlations can degrade dramatically on test data when a distribution shift occurs between the training/tuning data and test data, even though they demonstrate perfect performance on training/tuning data. In other words, models that employ spurious correlations exhibit poor out-of-distribution (OOD) generalization performance. A further complication is that this issue is especially preva-

lent in complex datasets where high-dimensional inputs, including image data and text data, may contain hidden biases.

Although considerable efforts have been made to mitigate spurious correlations in both visual modality (Arjovsky et al., 2019; Creager et al., 2021; Yang et al., 2023b; Qiu et al., 2024) and textual modality (Peyrard et al., 2022; Zhou et al., 2023), these methods are primarily designed for single-modal learning and are not applicable to multi-modal learning. In contrast to single-modal learning, the critical challenge of cross-modal mitigation of spurious correlations lies in ***how to organically integrate mitigation in visual modality, mitigation in textual modality and cross-modal alignment of representations***.

Among recent studies, the cross-modal contrastive learning framework presented in (Yang et al., 2023c) addresses the mitigation of spurious correlations in both textual and visual modalities while requiring access to text descriptions of spurious features/objects. In general scenarios, spurious features are typically latent and unobservable. Moreover, the method proposed in (Yang et al., 2023c), which is designed for fine-tuning of VLMs and alters all model parameters, cannot be applied to prompt tuning of VLMs. Besides, CoOPood (Zhang et al., 2024b) focuses on mitigating spurious correlations in visual modality during prompt-tuning of VLMs. It overlooks the spurious correlations in the textual modality. Furthermore, CoOPood relies on the assumption that the spurious correlations between spurious features and the target label are approximately subject to uniform probability distributions. Therefore, how to organically integrate mitigation in visual modality, mitigation in textual modality and cross-modal alignment of representations, without invoking unnatural assumptions, remains an open problem.

Inspired by the causal intervention-based calculation of the probability of necessity and sufficiency (PNS) between two variables (Tian & Pearl, 2000; Wang & Jordan, 2021; Yang et al., 2023b), we introduce the concept *logic alignment* (i.e., alignment with necessity and sufficiency) to integrate mitigation of spurious correlations and cross-modal alignment of representations organically for prompt tuning of VLMs. The key insight is that logic equivalence (i.e., necessary and sufficient) not only facilitates mitigation of spurious correlations (Wang & Jordan, 2021; Yang et al., 2023b), but also enhances dimensionality-agnostic alignment between two variables. In the context of vision-language models, the overall objective is to achieve the logic equivalence between visual causal representations (denoted by $\Phi_v$) and textual label (denoted by $Y$), i.e., $Y \Leftrightarrow \Phi_v$. Considering spurious correlations can exist in both visual and textual modalities, the equivalence $Y \Leftrightarrow \Phi_v$ alone cannot guarantee that the aligned textual representations exclude spurious features. Therefore, establishing a stricter equivalence chain $Y \Leftrightarrow \Phi_t \Leftrightarrow \Phi_v$ (where $\Phi_t$ represents textual causal representations) is our final objective. Specifically, our framework can be divided into two components: 1) $Y \Leftrightarrow \Phi_t$ eliminates the spurious correlations in textual modality; 2) $\Phi_t \Leftrightarrow \Phi_v$ integrates mitigation of spurious correlations in visual modality and cross-modal alignment of representations organically when $Y \Leftrightarrow \Phi_t$ excludes spurious features in $\Phi_t$. In practical implementation, the logic equivalence between two variables is achieved by maximizing the probability of necessity and sufficiency (PNS) between them. The main contributions of this work are summarized as follows:

- We introduce the concept *logic alignment* to address cross-modal mitigation of spurious correlations for prompt-tuning in vision-language models. Capable of integrating mitigation of spurious correlations and cross-modal alignment of representations organically, *Logic alignment* can serves as a promising technique for handling spurious correlations in various multi-modal learning scenarios.

- We design a practical framework to calculate the PNS between the textual label and textual representations, as well as the PNS between textual representations and visual representations. By maximizing these two PNS terms, the proposed objective can effectively achieve cross-modal mitigation of spurious correlations for prompt-tuning in VLMs.

- The theoretical analysis proves that our method can yield a tighter generalization error bound compared to existing approaches. Moreover, the detailed components of the derived generalization error bound verify the importance of maximizing the two proposed PNS terms from a theoretical perspective.

- The experimental results across diverse datasets demonstrate the superiority of the proposed framework in out-of-distribution generalization performance, compared with the state-of-the-art competitors.

## 2 RELATED WORK

**Causal Representation Learning** Attaining causally invariant predictors over varied data distributions is proposed in the field of causal inference Peters et al. (2016), and introduced into machine learning to tackle the OOD generalization problem by IRM Arjovsky et al. (2019). Then, many efforts are dedicated to facilitating the application of invariant representation learning to more general scenarios. Some works focus on achieving invariant learning when the environment label is unavailable, e.g., EIIL Creager et al. (2021), HRM Liu et al. (2021a), KerHRM Liu et al. (2021b), ED-NIL Huang et al. (2022) and ZIN Lin et al. (2022). IFM Chen et al. (2022b) lowers the requirement on the number of available environments. Another branch Ahuja et al. (2021); Chen et al. (2022a); Huh & Baidya (2022) completes the constraints that IRM misses. Besides, iCaRL Lu et al. (2022) extends causal representation learning to non-linear causal representations while ACTIR Jiang & Veitch (2022) extends causal representation learning to anti-causal scenarios. Causal representation learning is also applied to graph representation learning Li et al. (2022); Chen et al. (2022c) and natural language modeling Peyrard et al. (2022). These methods are devised for handling spurious correlations in single-modal learning scenarios.

**Prompt Tuning of Vision-Language Models** The typical vision-language model, CLIP (Radford et al., 2021) is trained using a contrastive learning framework where textual and visual representations are aligned by maximizing the cosine similarity between the image and text embeddings of correct pairs. To fully exploit the powerful adaptation capability, prompt tuning is proposed to improve the performance of pre-trained vision-language models (e.g., CLIP) on downstream task (Zhou et al., 2022b;a). Among these attempts, CoOp (Zhou et al., 2022b) designs learnable prompts to adjust the mapping from textual label to textual representations and greatly improves the performance of pre-trained CLIP on downstream visual tasks. Furthermore, CoCoOp (Zhou et al., 2022a) introduce a image-conditional context generator to improve the zero-shot generalization performance of CoOp. Subsequently, MaPLe (Khattak et al., 2023a) adopts both textual and visual learnable prompts to enhance the alignment of textual and visual representations in downstream tasks. Another prevalent line of works utilize fine-grained learnable textual prompt to tackle the imbalance between textual and visual modalities (Chen et al., 2023; Shen et al., 2024; Li et al., 2024). All above prompt tuning methods do not consider the mitigation of spurious correlations in vision-language models. In particular, CoOPood (Zhang et al., 2024b) is proposed as a pioneering work focusing on mitigating spurious correlations in visual modality during prompt-tuning of VLMs. However, it overlooks the spurious correlations in the textual modality. Moreover, CoOPood relies on the assumption that the spurious correlations between spurious features and the target label are approximately subject to uniform probability distributions, which limits the applicability of CoOPood to general scenarios.

## 3 PRELIMINARY

We introduce the background knowledge about prompt tuning of VLMs and causally motivated calculation for probability of necessity and sufficiency (i.e., PNS) in this section.

### 3.1 PROMPT TUNING OF CLIP

Contrastive Language-Image Pre-training (CLIP) (Radford et al., 2021) maintains two separate encoder: text encoder extracting textual representations from the text input and image encoder drawing visual representations from the image input. Textual and visual representations are aligned by conducting contrastive learning based on the language-image data pairs. For the sake of simplicity, we denote the text encoder as $f$ and image encoder as $g$ in CLIP. With a handcrafted prompt (e.g., a photo of a [CLASS]) input into the frozen text encoder, the pre-trained CLIP can be deployed to downstream image classification tasks. Specifically, input images are fed to the image encoder, while the text prompt is input into the text encoder. Suppose "[CLASS]" has $K$ categories in current downstream task, the pre-trained CLIP can make a probability prediction for input image $x$ by

$$p(k \mid x) = \frac{\exp(sim(z_t^k, g(x))/\tau)}{\sum_{j=1}^{K} \exp(sim(z_t^j, g(x))/\tau)} \quad (1)$$

where $z_t^j$, $j \in 1, 2, ..., K$ denotes text feature generated for class $j$ by the text encoder $f$. $sim(a, b)$ denotes the cosine similarity between two vector $a$ and $b$ while $\tau$ is the temperature parameter.

In order to improve the performance of pre-trained CLIP in downstream tasks, CoOp (Zhou et al., 2022b) introduces learnable text prompt to amend the mapping from text labels to textual representations. Suppose the learnable context is denoted as $Q = [q_1, q_2, ..., q_N]$, the complete text input can be written as $Q_C = [q_1, q_2, ..., q_N, \text{CLASS}]$. When the text input $Q_C = [Q, \text{CLASS}]$ is fed to the frozen text encoder, the corresponding textual feature vector for class $k$ can be written by $z_t^k = f([Q, k])$. For each instance $(x_i, y_j)$ in the tuning dataset $D_{\mathcal{S}} := \{(x_i, y_i)\}_{i=1}^m$, the model can provide a prediction by $p(y_i \mid x_i) = \frac{\exp(sim(f([Q, y_i]), g(x_i))/\tau)}{\sum_{j=1}^K \exp(sim(f([Q, j]), g(x_i))/\tau)}$. The learnable text prompt is optimized by solving the following objective:

$$\min_Q \mathcal{L}_{CE-logit} := - \sum_{(x_i, y_i) \in D_{\mathcal{S}}} y_i \log p(y_i \mid x_i). \tag{2}$$

Since only text prompt is learnable while both text and image encoder are frozen during the tuning stage, prompt tuning is a parameter-efficient tuning scheme and has gained great success.

## 3.2 PROBABILITY OF NECESSITY AND SUFFICIENCY (PNS)

Probability of Necessity and Sufficiency (PNS) describe the probability with which a variable is the necessary and sufficient cause of another variable. The formal definition of PNS is given as follows.

**Definition 3.1** (Probability of Necessity and Sufficiency (Pearl, 2009)). *Let the specific implementations of causal variable $\Phi$ as $\phi$ and $\bar{\phi}$, where $\phi \neq \bar{\phi}$. The probability with which variable $\Phi$ is the necessary and sufficient cause of variable $Y$ on test data distribution $P_{\mathcal{T}}$ is given by:*

$$PNS(Y, \Phi) := \underbrace{P_{\mathcal{T}}(Y_{do(\Phi=\phi)} = y \mid \Phi = \bar{\phi}, Y \neq y)}_{sufficiency} P_{\mathcal{T}}(\Phi = \bar{\phi}, Y \neq y)$$

$$+ \underbrace{P_{\mathcal{T}}(Y_{do(\Phi=\bar{\phi})} \neq y \mid \Phi = \phi, Y = y)}_{necessity} P_{\mathcal{T}}(\Phi = \phi, Y = y), \tag{3}$$

*where $do(\Phi = \phi)$ (do-operator) means the manipulable variable $\Phi$ is forced to be a fixed value $\phi$.*

Since the probability of necessity and sufficiency is defined based on counterfactual distributions, it is usually intractable to estimate the PNS of two variables. However, with two assumptions (Exogeneity and Monotonicity) proposed and utilized in (Pearl, 2009; Yang et al., 2023b), we can obtain a useful lemma as follows. Considering the limited length of main text, we put more detailed explanations about Exogeneity and Monotonicity assumption in Appendix C.

**Lemma 3.2** (Pearl (2009); Yang et al. (2023b)). *If variable $\Phi$ is exogenous relative to variable $Y$, and $Y$ is monotonic relative to $\Phi$, we can get*

$$PNS(Y, \Phi) = \underbrace{P_{\mathcal{T}}(Y = y \mid \Phi = \phi)}_{sufficiency} - \underbrace{P_{\mathcal{T}}(Y = y \mid \Phi = \bar{\phi})}_{necessity}. \tag{4}$$

## 3.3 PNS RISK MODELING

According to definition 3.1, PNS risk is based on the measure of $\phi$ and $\bar{\phi}$. As $\bar{\phi}$ represents the intervention value, it is not necessary for it to be a sample from the same distribution as the causal variable $\Phi$. Thus, we need an auxiliary variable $\bar{\Phi} \in \mathcal{Z}$ (within the same space as variable $\Phi$). The intervention value $\bar{\phi}$ is sampled from the distribution $P_{\mathcal{T}}(\bar{\Phi} \mid X = x)$. To calculate the probability of necessity and sufficiency between the representations and the target in neutral networks, we need to construct three networks parameterized by $\theta$ and $\xi$ to estimate the distributions $P_{\mathcal{T}}(\Phi \mid X = x) =$ and $P_{\mathcal{T}}(\bar{\Phi} \mid X = x)$ by $P_{\mathcal{T}}^{\theta}(\Phi \mid X = x) =$ and $P_{\mathcal{T}}^{\xi}(\bar{\Phi} \mid X = x)$, respectively. Additionally, we need to build a linear classifier $\omega$ to parameterize the mapping from causal representations to target. That is, the target can be obtained by $y = \text{sign}(\omega^{\intercal}\phi)$ (Yang et al., 2023b).

Let $\mathcal{I}(A)$ be an indicator function, where $\mathcal{I}(A) = 1$ if $A$ is true; otherwise, $\mathcal{I}(A) = 0$. PNS risk based on Definition 3.1 and Lemma 3.2 can be calculated by

$$\mathcal{R}_{\mathcal{S}}(\omega, \theta, \xi) := \mathbb{E}_{(x,y) \sim D_{\mathcal{S}}} \left[ \mathbb{E}_{\phi \sim P_{\mathcal{S}}(\Phi|X=x)} \mathcal{I}[\text{sign}(\omega^{\intercal}\phi) \neq y] + \mathbb{E}_{\bar{\phi} \sim P_{\mathcal{S}}(\bar{\Phi}|X=x)} \mathcal{I}[\text{sign}(\omega^{\intercal}\bar{\phi}) = y] \right] \tag{5}$$

For practical modeling convenience, a recent study (Yang et al., 2023b) proposed an effective approximation scheme for PNS risk by deriving an upper bound of Equation 5.

**Proposition 3.3** (Proposition 3.1 in (Yang et al., 2023b)). *Given a source domain $\mathcal{S}$, we define the sufficient and necessary risks as:*

$$SF_{\mathcal{S}}(\omega, \theta) := \mathbb{E}_{(x,y) \sim D_{\mathcal{S}}} \mathbb{E}_{\phi \sim P_{\mathcal{S}}^{\theta}(\Phi|X=x)} \mathcal{I}[\text{sign}(\omega^{\intercal}\phi) \neq y],$$

$$NC_{\mathcal{S}}(\omega, \xi) := \mathbb{E}_{(x,y) \sim D_{\mathcal{S}}} \mathbb{E}_{\bar{\phi} \sim P_{\mathcal{S}}^{\xi}(\bar{\Phi}|X=x)} \mathcal{I}[\text{sign}(\omega^{\intercal}\bar{\phi}) = y],$$

*and let the Monotonicity measurement be defined as*

$$M_{\mathcal{S}}^{\omega}(\theta, \xi) := \mathbb{E}_{(x,y) \sim D_{\mathcal{S}}} \mathbb{E}_{\phi \sim P_{\mathcal{S}}^{\theta}(\Phi|X=x)} \mathbb{E}_{\bar{\phi} \sim P_{\mathcal{S}}^{\xi}(\bar{\Phi}|X=x)} \mathcal{I}[\text{sign}(\omega^{\intercal}\phi) = \text{sign}(\omega^{\intercal}\bar{\phi})],$$

*then we have*

$$\mathcal{R}_{\mathcal{S}}(\omega, \theta, \xi) = M_{\mathcal{S}}^{\omega}(\theta, \xi) + 2SF_{\mathcal{S}}(\omega, \theta)NC_{\mathcal{S}}(\omega, \xi) \leq M_{\mathcal{S}}^{\omega}(\theta, \xi) + 2SF_{\mathcal{S}}(\omega, \theta). \quad (6)$$

Based on the upper bound derived in Proposition 3.3, CaSN (Yang et al., 2023b) maximizes the PNS between variable $\Phi$ and variable $Y$ by solving the following optimization problem:

$$\min_{\omega, \theta} \max_{\xi} \mathcal{L}_{PNS}(\omega, \theta, \xi) := M_{\mathcal{S}}^{\omega}(\theta, \xi) + SF_{\mathcal{S}}(\omega, \theta) + \lambda \mathcal{R}_{KL}, \quad \text{subject to} \quad \|\phi - \bar{\phi}\| \geq \delta, \quad (7)$$

where $\mathcal{R}_{KL} := \mathbb{E}_{D_{\mathcal{S}}} KL(P_{\mathcal{S}}^{\theta}(\Phi \mid X = x) \| \pi_{\Phi}) + \mathbb{E}_{D_{\mathcal{S}}} KL(P_{\mathcal{S}}^{\xi}(\bar{\Phi} \mid X = x) \| \pi_{\bar{\Phi}})$. $KL(\cdot, \cdot)$ denotes the KL-divergence between two probability distributions. $\pi_{\Phi} := P_{\mathcal{S}}(\Phi)$ and $\pi_{\bar{\Phi}} := P_{\mathcal{S}}(\bar{\Phi})$ describe the prior distributions of $\Phi$ and $\bar{\Phi}$, respectively.

## 4 METHODOLOGY

In this section, we first discuss the detailed design of the proposed framework LogicAl-PT in Section 4.1 and then provide theoretical analysis on generalization error bound to demonstrate the effectiveness of the proposed method from the theoretical perspective in chapter 4.2.

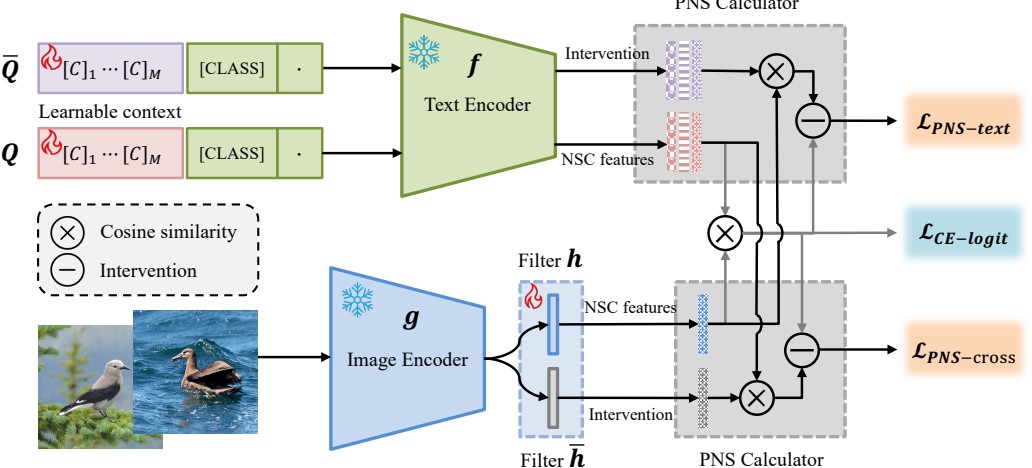

Figure 1: Overall framework of LogicAl-PT. "NSC" represents "necessary and sufficient cause". Two filters behind the image encoder are implemented using two linear layer, respectively. The NSC features in textual and visual modalities are given by $f([Q, \text{CLASS}])$ and $h(g(X))$, respectively. The interventions in textual and visual modalities are given by $f([\bar{Q}, \text{CLASS}])$ and $\bar{h}(g(X))$, respectively. Only $f([Q, \text{CLASS}])$ and $h(g(X))$ are utilized for predicting at inference phase.

### 4.1 OVERVIEW OF LOGICAL-PT

In order to achieve effective cross-modal mitigation of spurious correlations for prompt-tuning in vision-language models, we design a practical framework which can be divided into two components: 1) $Y \Leftrightarrow \Phi_t$ eliminates the spurious correlations and enhances logic alignment in textual modality; 2) $\Phi_t \Leftrightarrow \Phi_v$ integrates mitigation of spurious correlations in visual modality and cross-modal alignment of representations organically when $Y \Leftrightarrow \Phi_t$ excludes spurious features in $\Phi_t$. The overall framework of the proposed method LogicAl-PT is displayed in Figure 1.

**Cross-modal logic alignment.** As shown in objective (7), constructing the parameterized mapping $\omega$, $\theta$ and $\xi$ is necessary for calculating the PNS risk. When we aim at achieving cross-modal logic alignment, we need to maximize the probability of necessity and sufficiency between visual representations and textual representations. In the design framework, two filters $h$ and $\bar{h}$ serve as the parameterized mapping $\theta$ and $\xi$, respectively. Moreover, $f([Q, \text{CLASS}])$ can work as the classifier $\omega$. Therefore, the PNS risk corresponding to cross-modal logic alignment is given by

$$\mathcal{L}_{PNS-cross} = \mathcal{L}_{PNS}(f([Q, \text{CLASS}]), h, \bar{h}). \tag{8}$$

**Textual logic alignment.** When we calculate textual PNS risk to achieve textual logic alignment, $f([Q, \text{CLASS}])$ and $f([\bar{Q}, \text{CLASS}])$ serve as the parameterized mapping $\theta$ and $\xi$, respectively. To construct the classifier $\omega$ for textual representations, we draw the prototype of each class from the visual representation space $h(g(X))$. These prototypes can serve as a classifier for the textual representations by calculating cosine similarity-based logit. In this way, the PNS risk corresponding to textual logic alignment is given by

$$\mathcal{L}_{PNS-text} = \mathcal{L}_{PNS}(h(g(X)), f([Q, \text{CLASS}]), f([\bar{Q}, \text{CLASS}])). \tag{9}$$

**Overall objective.** As shown in Figure 1, the cross-modal cross-entropy loss $\mathcal{L}_{CE-logit}$ is computed utilizing the cosine similarity between textual representations $f([Q, \text{CLASS}])$ and visual representations $h(g(X))$. Therefore, the overall train objective can be written as:

$$\min_{Q,h} \max_{\bar{Q},\bar{h}} \mathcal{L}_{CE-logit} + \alpha \mathcal{L}_{PNS}(f([Q, \text{CLASS}]), h, \bar{h}) + \beta \mathcal{L}_{PNS}(h(g(X)), f([Q, \text{CLASS}]), f([\bar{Q}, \text{CLASS}])). \tag{10}$$

During the inference phase, the probability prediction for an input image is calculated by using the cosine similarity between textual and visual "NSC" features, i.e., $f([Q, \text{CLASS}])$ and $h(g(X))$.

## 4.2 THEORETICAL ANALYSIS

Along the information flow from visual representations $\Phi_v$ to text label $Y$ in a vision-language model, we can evaluate the effectiveness of the visual feature extractor $\Phi_v$ in predicting the target $Y$ using the mutual information $I(Y; \Phi_v(X))$. In practice, we can acquire the empirical estimation of $I(Y; \Phi_v(X))$ on the source dataset $D_{\mathcal{S}}$, represented as $\hat{I}_{\mathcal{S}}(Y; \Phi_v(X))$. When the learning model is ready for deployment, we prioritize the performance of $\Phi_v$ on some unknown target data distribution, denoted by $I_{\mathcal{T}}(Y; \Phi_v(X))$. Since $I_{\mathcal{T}}(Y; \Phi_v(X))$ is inaccessible, bounding the generalization error $I_{\mathcal{T}}(Y; \Phi_v(X)) - \hat{I}_{\mathcal{S}}(Y; \Phi_v(X))$ is critical for analysing the generalization performance of the proposed method in learning theory.

Before starting to the theoretical analysis on generalization error bound, we first introduce a useful assumption for the following theoretical analysis.

**Assumption 4.1.** *In the textual modal, the textual representations $\Phi_t$ are fully informative for determining the target $Y$. That is, we have $Y \perp\!\!\!\perp \Phi_v \mid \Phi_t$.*

**Theorem 4.2.** *Suppose the source and target data distributions are denoted by $\mathbb{P}_{\mathcal{S}}(X, Y)$ and $\mathbb{P}_{\mathcal{T}}(X, Y)$, respectively, and the size of the source dataset $D$ is $m$. Then, there exists a finite constant $C$ such that the following inequality holds with a probability at least $1 - \delta$:*

$$\left| I_{\mathcal{T}}(Y; \Phi_v(X)) - \hat{I}_{\mathcal{S}}(Y; \Phi_v(X)) \right| \leq \underbrace{\frac{\sqrt{C \log(|\mathcal{Y}|/\delta)}\Big(|\mathcal{X}| \log(m) + |\mathcal{Y}| \log(|\mathcal{Z}|)\Big) + \frac{2}{e}|\mathcal{X}|}{\sqrt{m}}}_{\textit{Empirical error term}}$$

$$+ \underbrace{\mathcal{J}(Y|\Phi_t) + \sqrt{C|\mathcal{Y}|\mathcal{J}(Y|\Phi_t)}}_{\textit{Textual error term}} + \underbrace{\mathcal{J}(\Phi_t|\Phi_v) + \sqrt{C|\mathcal{Y}|\mathcal{J}(\Phi_t|\Phi_v)}}_{\textit{Alignment error term}},$$

*where $m \geq \frac{C}{4} \log(|\mathcal{Y}|/\delta)|\mathcal{X}|e^2$. The term 'Textual error term' is caused by distribution shift in textual modality while 'Alignment error term' stems from the misalignment between textual and visual modalities. $\mathcal{J}(Y|\Phi_t)$ denotes the Jeffrey's divergence defined by*

$$\mathcal{J}(Y|\Phi_t) \triangleq \mathcal{KL}\big(\mathbb{P}_{\mathcal{T}}(Y \mid \Phi_t)\|\mathbb{P}_{\mathcal{S}}(Y \mid \Phi_t)\big) + \mathcal{KL}\big(\mathbb{P}_{\mathcal{S}}(Y \mid \Phi_t)\|\mathbb{P}_{\mathcal{T}}(Y \mid \Phi_t)\big)$$

*where $\mathcal{KL}(\cdot\|\cdot)$ denotes the Kullback–Leibler divergence between two probability distributions. Similarly, the term $\mathcal{J}(\Phi_t|\Phi_v)$ is given be*

$$\mathcal{J}(\Phi_t|\Phi_v) \triangleq \mathcal{KL}\big(\mathbb{P}_{\mathcal{T}}(\Phi_t \mid \Phi_v(X))\|\mathbb{P}_{\mathcal{S}}(\Phi_t \mid \Phi_v(X))\big) + \mathcal{KL}\big(\mathbb{P}_{\mathcal{S}}(\Phi_t \mid \Phi_v(X))\|\mathbb{P}_{\mathcal{T}}(\Phi_t \mid \Phi_v(X))\big).$$

**Remark 4.3.** *The first term 'Empirical error term' stems from limited number of data samples and will approach $0$ as the size of source dataset grows towards infinity. As regard to the second term 'Textual error term' caused by spurious correlations in textual modality, it can be unbounded and equals to $0$ if and only if $\mathbb{P}_\mathcal{T}(Y|\Phi_t) = \mathbb{P}_\mathcal{S}(Y|\Phi_t)$. When the textual representations encode spurious correlations, the second term is always strictly larger than $0$. As comparison, the third term 'Alignment error term' is caused by the misalignment between textual and visual representations. Similarly, the 'Alignment error term' is always non-negative and equals $0$ if and only if $\mathbb{P}_\mathcal{T}(\Phi_t|\Phi_v) = \mathbb{P}_\mathcal{S}(\Phi_t|\Phi_v)$. According to the results in Theorem 4.3 in (Yang et al., 2023b), we know that optimizing the PNS risk in equation 8 can guarantee $Y \perp\!\!\!\perp Q \mid \Phi_t$ and optimizing the PNS risk in equation 9 can enable $\Phi_t \perp\!\!\!\perp X \mid \Phi_v$. Therefore, the proposed method can render both 'Textual error term' and 'Alignment error term' approach $0$. In other words, our method can guarantee a tighter generalization error bound compared with the state-of-the-art prompt-tuning schemes for vision-language models. Detailed proof of Theorem 4.2 is provided in Appendix B.*

## 5 EXPERIMENTS

### 5.1 EXPERIMENTAL SETUP

**Datasets** To evaluate the performance of the proposed LogicAl-PT, we conduct experiments on four commonly used datasets: Waterbird (Sagawa et al., 2019), CelebA (Liu et al., 2015), ImageNet-1K (Russakovsky et al., 2015), and PACS Li et al. (2017). Detailed setup is explained as follows.

Waterbirds is a commonly used benchmark dataset for studying spurious correlations. The task is to classify whether an image shows a landbird or a waterbird. The background (land and water) serve as a spurious attribute for classification of bird images. Images in Waterbird dataset can be divided into four groups: landbirds on land background (G1), landbirds on water background (G2), waterbirds on land background (G3) and waterbirds on water background (G4). The number of pictures within these four groups account for 73.0%, 3.8%, 1.2%, and 22.0% of the data, respectively. Group G3 is the minority group. In the training set, landbirds appeared more often on land backgrounds, while waterbirds appeared more often on water backgrounds, so models fine-tuned on this dataset tended to rely on backgrounds rather than birds to make prediction. However, in the testing set, both landbirds and waterbirds have the same probability of appearing on a land background as on a water background, which leads to a degradation of the model's performance.

Similar to Watebirds, CelebA is a hair color prediction dataset, which also has 4 groups: non-blond females (G1), non-blond males (G2), blond females (G3) and blond males (G4) with proportions 3.9%, 73.9%, 21.1%, and 1.1% of the data, respectively. Group G4 is the minority group.

In ImageNet-1K, there are features spuriously correlated with some categories (Singla et al., 2021). For example, for Baby pacifier class, the spurious attribute is baby face. Samples without babies in the image are susceptible to being classified as water bottles rather than baby pacifier. CLIP using ResNet-50 has a 98.2% classification accuracy for samples with babies in the image, but only 36.1% for samples without babies. We use the water bottle class and the baby pacifier class in ImageNet-1K as the training set, which has three groups: water bottles (G1), baby pacifier without baby (G2), baby pacifier with baby (G3), accounting for 73.9%, 5.2%, and 20.9% of the data, respectively; the group G2 is the minority group. Note that since the validation set for ImageNet contains only 50 images per class, we transferred a portion of the data from the original training set to the test set.

PACS is a larger real-world dataset commonly used for evaluating out-of-distribution (OOD) generalization. It consists of 7 classes distributed across 4 domains. We adopt the "leave-one-domain-out" strategy to evaluate OOD generalization performance. For example, when evaluating performance on 'Art Painting' domain, the remaining three domains are used as train domains.

**Baseline Methods.** We compare the performance of our LogicAl-PT with the state-of-the-art competitors, including the zero-shot CLIP (Radford et al., 2021); CoOp (Zhou et al., 2022b), a widely adopted prompt tuning method, which only minimize the contrastive loss $\mathcal{L}_{CE-logit}$; Empirical Risk Mimimization (ERM), the standard technique for minimizing classification loss which also only minimize the cross-entropy loss; and CoOPood (Zhang et al., 2024b) which aligns the textual representations with the decoupled invariant representations. It is noted that, different from CoOp, under our model framework, the ERM method will use the causal projection layer (i.e., $h$ in Fig-

ure 1). Besides, we also introduce two state-of-the-art prompt tuning methods as competitors: 1) PromptSRC (Khattak et al., 2023b) which designs a self-regulating framework for prompt learning and DePT (Zhang et al., 2024a) which decouples the base-specific knowledge from feature channels into an isolated feature space during prompt tuning of VLMs.

## 5.2 OVERALL PERFORMANCE

Table 1: Overall performance comparison among LogicAl-PT and the state-of-the-art competitors.

| Backbones | ResNet-50 | | | | | | | | ViT-B/32 | | | | | | | |
|---|---|---|---|---|---|---|---|---|---|---|---|---|---|---|---|---|
| Datasets | Waterbird | | CelebA | | ImageNet | | PACS | | Waterbird | | CelebA | | ImageNet | | PACS | |
| Test Acc (%) | Worst | Avg | Worst | Avg | Worst | Avg | Worst | Avg | Worst | Avg | Worst | Avg | Worst | Avg | Worst | Avg |
| CLIP | 43.6 | 70.7 | 67.8 | 84.1 | 36.6 | 68.2 | 80.2 | 91.5 | 41.4 | 65.3 | 69.7 | 85.2 | 51.4 | 75.8 | 81.7 | 93.8 |
| CoOp | 49.3 | 79.1 | 28.9 | 80.6 | 77.3 | 87.7 | 81.3 | 92.4 | 43.5 | 77.4 | 26.2 | 77.0 | 87.1 | 92.8 | 82.4 | 94.5 |
| ERM | 54.7 | 84.1 | 26.7 | 78.2 | 80.5 | 88.5 | 80.0 | 92.6 | 49.6 | 78.3 | 25.9 | 76.8 | 86.7 | 93.3 | 82.9 | 94.1 |
| CoOPood | 60.3 | **86.3** | 31.6 | 78.6 | 85.8 | 92.9 | 81.5 | 92.8 | 52.5 | 79.2 | 27.1 | 76.5 | 89.9 | 94.6 | 82.7 | 94.4 |
| PromptSRC | 57.2 | 85.5 | 68.2 | 85.3 | 81.6 | 89.4 | 81.7 | 93.6 | 50.8 | 79.5 | 69.3 | 85.9 | 87.8 | 94.1 | 83.4 | 94.8 |
| DePT+PromptSRC | 57.9 | 86.0 | 68.3 | 85.7 | 82.0 | 90.1 | 81.6 | **93.9** | 51.7 | 80.0 | 70.2 | 86.3 | 87.4 | 94.3 | 83.5 | 95.1 |
| LogicAl-PT | **67.5** | 86.2 | **69.9** | **87.3** | **90.2** | **95.1** | **82.4** | 93.7 | **61.2** | **80.3** | **73.1** | **86.9** | **91.8** | **95.4** | **84.3** | **95.2** |

To assess OOD generalization performance, we evaluate the test accuracy of the obtained models across a range of diverse test data distributions (4 test domains in Waterbird, CelebA, 3 test domains in ImageNet-1K dataset, and 4 test distributions in PACS). Among them, the worst-case (Worst) accuracy and average (Avg) accuracy are summarized in Table 1. Since the test data distribution is unknown in practical scenarios, both the worst-case and average accuracy are significant for reflecting the OOD generalization performance of a model. As shown in Table 1, our method LogicAl-PT outperforms the competitors on both worst-case and average test accuracy in four commonly used datasets. In particular, LogicAl-PT achieves around 7% / 9%, 2% / 3%, 4% / 2% and 1% / 1% higher worst-case accuracy than the second best algorithm on Waterbird, CelebA, ImageNet-1K and PACS when ResNet-50 / ViT-B/32 is used as backbone model, respectively.

## 5.3 VISUALIZATION

For the purpose of verifying that the tuned models developed by our method LogicAl-PT exploit the necessary and sufficient features rather than spurious features, we sample some data instances to generate visual explanations for the selected model using Grad-CAM (Selvaraju et al., 2017). The commonly used Grad-CAM can produce a localization map which highlights the important regions in the input image that a deep learning model depends on for predicting the label. As shown in Figure 2, the pivotal features employed by various prompt tuning methods and zero-shaot CLIP for predicting WaterBird (Figure 2(a)) and BabyPacifier (Figure 2(b)) are highlighted in red.

The visualization results reveal that the proposed LogicAl-PT demonstrates three notable advantages over existing prompt-tuning methods: **1) LogicAl-PT can effectively eliminate the non-causal spurious features** that are associated with the label (i.e., 'background' in WaterBird dataset and 'baby' in ImageNet-1K dataset). **2) LogicAl-PT can mitigate the 'sufficient but not necessary' features** that demonstrate inconsistent presence across different data instances. For example, the shape of feet is a 'sufficient but not necessary' feature for classifying the picture of a bird as 'waterbird' or 'landbird' because its feet can retract or remain hidden when the bird is lying down or in flight. **3)** As shown in Figure 2(a), **LogicAl-PT can mitigate the 'necessary but not sufficient' features** which can impact the classification performance when the distribution of these 'necessary but not sufficient' features varies. For example, the wings of birds are 'necessary but not sufficient' features for distinguishing 'waterbird' from 'landbird'. From the visualization results in Figure 2(a), we can find that LogicAl-PT avoids utilizing the wings to categorize the pictures of birds.

In summary, visualization results demonstrate the proposed LogicAl-PT can effectively exploit the 'sufficient and necessary' features and mitigate the unstable features, including non-causal spurious features, 'sufficient but not necessary' features and 'necessary but not sufficient' features. This explains why LogicAl-PT achieves superior out-of-distribution generalization performance, delivering more consistent results across diverse data distributions compared to its competitors.

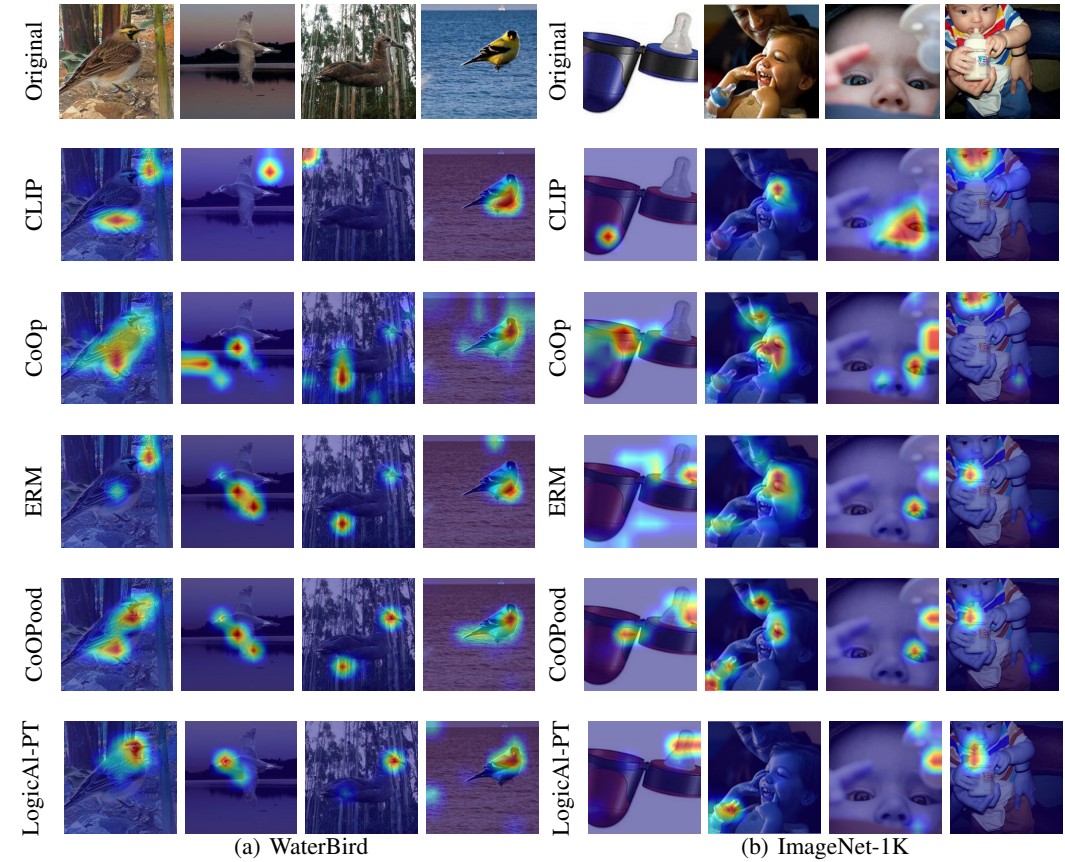

(a) WaterBird                          (b) ImageNet-1K

Figure 2: Visualization results of various prompt tuning approaches and zero-shot CLIP when predicting in WaterBird and ImageNet-1K datasets are generated by using Grad-CAM.

### 5.4 ABLATION STUDY

Table 2: The effect of the two separate regularization terms in the overall objective.

| Datasets | Waterbird | | CelebA | | ImageNet-1K | | PACS | |
|---|---|---|---|---|---|---|---|---|
| Test Acc (%) | Worst | Avg | Worst | Avg | Worst | Avg | Worst | Avg |
| LogicAl-PT ($\alpha = 0$) | 51.56 | 78.24 | 30.17 | 79.32 | 78.59 | 87.23 | 80.66 | 92.18 |
| LogicAl-PT ($\beta = 0$) | 65.45 | 85.72 | 67.51 | 85.74 | 88.64 | 94.31 | 80.75 | 93.27 |
| LogicAl-PT | **67.52** | **86.23** | **69.85** | **87.31** | **90.24** | **95.12** | **82.41** | **93.65** |

**Effect of Logic Alignments**   As discussed in Section 4.1, there are two significant regularization terms corresponding to the cross-modal logic alignment and textual logic alignment in the proposed optimization objective 10. We evaluate the isolated effects of them by independently setting $\alpha = 0$ and $\beta = 0$ in the objective 10, respectively. As displayed in Table 2, the results indicate that the cross-modal logic alignment is more important for cross-modal mitigation of spurious correlations than textual logic alignment. However, combining textual logic alignment with cross-modal logic alignment can further improve the out-of-distribution generalization performance. In this case, a natural question arises: '***Is textual alignment necessary, and what role does it serve during prompt tuning?***' We assess the necessity of textual logic alignment in the following paragraph.

**Necessity of Textual Logic Alignment**   Before studying the role textual logic alignment serves through the lens of visualization, we start from a qualitative analysis. Since the textual representations (corresponding to variable $\Phi_t$) are the class-wise mapping from the text labels, the sufficiency of variable $Y$ for variable $\Phi_t$ (i.e., $Y \Rightarrow \Phi_t$) is naturally guaranteed while the reverse $Y \Leftarrow \Phi_t$ is not

ensured. In other words, textual representations ($\Phi_t$) must be necessary causes for variable $Y$, but they don't have to be sufficient causes for variable $Y$. Therefore, textual logic alignment is proposed to enhance the sufficiency of text representations ($\Phi_t$) for label $Y$. Accordingly, when cross-modal logic alignment (i.e., $\Phi_t \Leftrightarrow \Phi_v$) is achieved, combining textual logic alignment can mitigate the visual features that are not sufficient for variable $Y$.

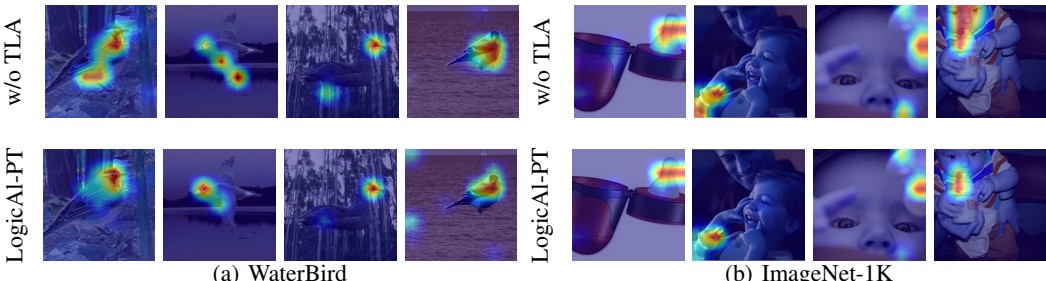

(a) WaterBird  (b) ImageNet-1K

Figure 3: Visualization results for assessing the necessity of textual logic alignment.

To investigate the actual role that textual logic alignment serves, we visualize the features which is utilized by the model tuned without textual logic alignment (w/o TLA), i.e., $\beta = 0$. In particular, when we set $\beta = 0$, $\alpha$ is tuned to its optimal value, i.e., the cross-modal logic alignment ($\Phi_t \Leftrightarrow \Phi_v$) is enhanced. The visualization results are displayed in Figure 3. Comparing the results, we can find that adding textual logic alignment can mitigate the visual features which are not sufficient for predicting $Y$. For example, adopting textual logic alignment mitigates the 'background' feature (on 3rd picture in Figure 3(a)) and 'wing' feature (on 2nd picture in Figure 3(a)) which are not sufficient features for making classification in WaterBird dataset, and mitigate the 'bottle' feature (on 2nd picture in Figure 3(b)) and 'baby face' feature (on 4th picture in Figure 3(b)) that are not sufficient features for predicting 'babypacifier' in ImageNet dataset. Therefore, we can conclude that the visualization results support the qualitative analysis.

Table 3: Performance of LogicAl-PT with different values of $\alpha$ and $\beta$ on ImageNet-1K.

| $\alpha$ | 0.0 | 1.0 | 10.0 | 20.0 | 30.0 | 50.0 |
|---|---|---|---|---|---|---|
| worst-case (%) | 78.6 | 80.9 | 86.1 | 90.2 | 88.7 | 79.5 |
| average (%) | 87.2 | 89.4 | 93.5 | 95.1 | 94.0 | 87.9 |
| $\beta$ | 0.0 | 0.10 | 1.00 | 10.0 | 20.0 | 30.0 |
| worst-case (%) | 88.6 | 89.7 | 90.2 | 89.3 | 87.2 | 85.5 |
| average (%) | 94.3 | 94.8 | 95.1 | 94.5 | 93.2 | 91.9 |

**Sensitivity of Hyper-parameters**    We evaluate the effects of two significant hyper-parameters in the proposed objective (i.e., $\alpha$ and $\beta$) on model performance here. Since the results on other datasets present the similar tendency as on ImageNet, we herein focus on ImageNet. When evaluating the effect of $\alpha$, we fix $\beta = 1.0$. When evaluating the effect of $\alpha$, we fix $\alpha = 20.0$. The results are shown in Table 3. We can find the performance of LogicAl-PT is more sensitive to the selection of $\alpha$ than the selection of $\beta$. To effectively mitigate spurious correlations in VLMs, careful tuning of $\alpha$ is essential. Regarding $\beta$, a small value is safer in practice, as a large $\beta$ may compromise the discriminative capability of the extracted features.

## 6    CONCLUSION

This paper investigates the cross-modal mitigation of spurious correlations in prompt tuning of vision-language models. We exploit causally motivated *logic alignment* (i.e., alignment with necessity and sufficiency) to integrate mitigation of spurious correlations and cross-modal alignment of representations organically. Theoretical analysis is provided to prove that our method can yield a tighter generalization error bound than existing approaches. Experimental results across diverse datasets demonstrate the superiority of the proposed framework, termed LogicAl-PT, in out-of-distribution generalization performance, compared with the state-of-the-art competitors.

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

# A MOTIVATION FOR UTILIZING PNS

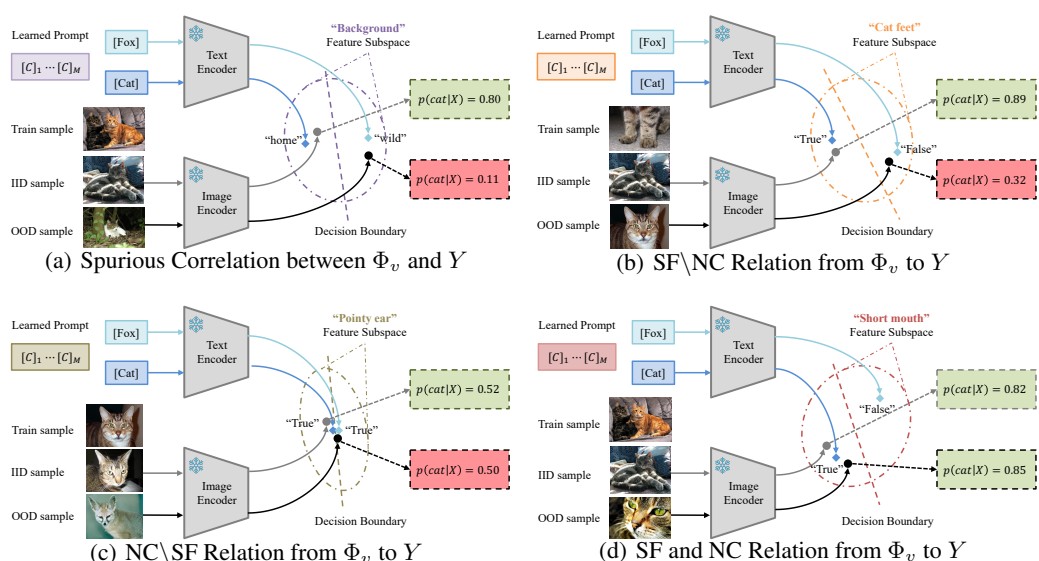

(a) Spurious Correlation between $\Phi_v$ and $Y$

(b) SF\NC Relation from $\Phi_v$ to $Y$

(c) NC\SF Relation from $\Phi_v$ to $Y$

(d) SF and NC Relation from $\Phi_v$ to $Y$

Figure 4: Illustration for three possible relations that are unstable across diverse data distributions (non-causal spurious correlation, SF\NC relation, and NC\SF relation) in vision-language models, where 'SF\NC' denotes 'sufficient but not necessary' and 'NC\SF' indicates 'necessary but not sufficient'. Besides, 'SF and NC' means 'sufficient and necessary' in Figure 4(d). 'IID' indicates 'in-distribution' while 'OOD' means 'out-of-distribution'. $\Phi_v$ represents the visual representation while $Y$ indicates text label.

In these examples, the task is a binary classification problem aimed at distinguishing 'cat' class from 'fox' class. The learned prompt together with the frozen text encoder works as a projector which projects the two text labels onto a specific feature subspace. When the text labels are projected into the 'Background' feature subspace (as shown in Figure 4(a)), the 'background' feature component in visual representation space determines the prediction result because prediction is made using cosine similarity between visual features and text features. In this way, a spurious correlation between visual representation and text label is built by this learned prompt. Similarly, the learned prompt in Figure 4(b) builds a SF\NC relation from $\Phi_v$ to $Y$, since 'cat feet' is a sufficient but not necessary feature for predicting 'cat'; the learned prompt in Figure 4(c) builds a NC\SF relation from $\Phi_v$ to $Y$, since 'pointy ear' is a necessary but not sufficient feature for predicting 'cat'; the learned prompt in Figure 4(d) builds a SF and NC relation (i.e., logic alignment) from $\Phi_v$ to $Y$, since 'short mouth' is a sufficient and necessary feature for predicting 'cat'.

As illustrated in Figure 4(a), 4(b) and 4(c), all these three relations (non-causal spurious correlation, SF\NC causal relation, and NC\SF causal relation) are unstable when data distribution varies. Therefore, apart from mitigation of cross-modal spurious correlations, cross-modal logic alignment (i.e., sufficiency and necessary) is also essential for enhancing the out-of-distribution generalization performance in vision-language models. This is why we utilize PNS risk in the prompt tuning of VLMs to achieve better out-of-distribution generalization performance.

# B THEORETICAL PROOF: GENERALIZATION ERROR BOUND

In this paper, we denote the true data distribution of source and target datasets as $p_{\mathcal{S}}$ and $p_{\mathcal{T}}$, respectively. In practical scenarios, the number of available data instances in a specific dataset is limited. We describe the empirical data distributions estimated from the source dataset and target dataset by $\hat{p}_{\mathcal{S}}$ and $\hat{p}_{\mathcal{T}}$, respectively. Without loss of generality, we use notations with subscripts $\mathcal{S}$ and $\mathcal{T}$ to represent metrics on the source and target data, respectively, while notations with the overscript ˆ denote empirical estimates (e.g., the empirical distribution $\hat{p}$ and the true distribution $p$).

**Proposition B.1** (Lemma 11 Shamir et al. (2010)). *Let $p$ be a distribution vector of arbitrary (possible countably infinite) cardinality, and $\hat{p}$ be an empirical estimation of $p$ based on a dataset of size $m$. Then with a probability of at least $1 - \delta$ over the samples, the following inequality holds:*

$$\|p - \hat{p}\| \leq \frac{2 + \sqrt{2\log(1/\delta)}}{\sqrt{m}} \tag{11}$$

**Theorem 4.7.** Suppose the source and target data distributions are denoted by $\mathbb{P}_{\mathcal{S}}(X, Y)$ and $\mathbb{P}_{\mathcal{T}}(X, Y)$, respectively, and the size of the source dataset $D$ is $m$. Then, there exists a finite constant $C$ such that the following inequality holds with a probability at least $1 - \delta$:

$$\left| I_{\mathcal{T}}(Y; \Phi_v(X)) - \hat{I}_{\mathcal{S}}(Y; \Phi_v(X)) \right| \leq \underbrace{\frac{\sqrt{C\log(|\mathcal{Y}|/\delta)}\Big(|\mathcal{X}|\log(m) + |\mathcal{Y}|\log(|\mathcal{Z}|)\Big) + \frac{2}{e}|\mathcal{X}|}{\sqrt{m}}}_{\textbf{Empirical error term}}$$

$$+ \underbrace{\mathcal{J}(Y|\Phi_t) + \sqrt{C|\mathcal{Y}|\mathcal{J}(Y|\Phi_t)}}_{\textbf{Textual error term}} + \underbrace{\mathcal{J}(\Phi_t|\Phi_v) + \sqrt{C|\mathcal{Y}|\mathcal{J}(\Phi_t|\Phi_v)}}_{\textbf{Alignment error term}},$$

where $m \geq \frac{C}{4}\log(|\mathcal{Y}|/\delta)|\mathcal{X}|e^2$. The term 'Textual error term' is caused by distribution shift in textual modality while 'Alignment error term' stems from the misalignment between textual and visual modalities. $\mathcal{J}(Y|\Phi_t)$ denotes the Jeffrey's divergence defined by

$$\mathcal{J}(Y|\Phi_t) \triangleq \mathcal{KL}\big(\mathbb{P}_{\mathcal{T}}(Y \mid \Phi_t)\|\mathbb{P}_{\mathcal{S}}(Y \mid \Phi_t)\big) + \mathcal{KL}\big(\mathbb{P}_{\mathcal{S}}(Y \mid \Phi_t)\|\mathbb{P}_{\mathcal{T}}(Y \mid \Phi_t)\big)$$

where $\mathcal{KL}(\cdot\|\cdot)$ denotes the Kullback–Leibler divergence between two probability distributions. Similarly, $\mathcal{J}(\Phi_t|\Phi_v)$ is given be

$$\mathcal{J}(\Phi_t|\Phi_v) \triangleq \mathcal{KL}\big(\mathbb{P}_{\mathcal{T}}(\Phi_t \mid \Phi_v(X))\|\mathbb{P}_{\mathcal{S}}(\Phi_t \mid \Phi_v(X))\big) + \mathcal{KL}\big(\mathbb{P}_{\mathcal{S}}(\Phi_t \mid \Phi_v(X))\|\mathbb{P}_{\mathcal{T}}(\Phi_t \mid \Phi_v(X))\big).$$

*Proof.* At the beginning of the proof, we denote the mutual information between $X$ and $Y$ which is computed on data distribution $\hat{p}_{\mathcal{S}}, \hat{p}_{\mathcal{T}}, p_{\mathcal{S}}$ and $p_{\mathcal{T}}$ by $\hat{I}_{\mathcal{S}}(Y; X), \hat{I}_{\mathcal{T}}(Y; X), I_{\mathcal{S}}(Y; X)$ and $I_{\mathcal{T}}(Y; X)$, respectively. We will derive the generalization error bound using the similar schemes as in (Shamir et al., 2010; Yang et al., 2023a; Tang et al., 2024).

Before starting the process of proof, we define a useful real-valued function $\xi$ as follows:

$$\xi(x) = \begin{cases} 0, & x = 0 \\ x\log(\frac{1}{x}), & 0 < x \leq \frac{1}{e} \\ \frac{1}{e}, & x > \frac{1}{e} \end{cases} . \tag{12}$$

It is noted that $\xi(x)$ is a continuous, monotonically increasing and concave real-valued function.

In general, we consider a deterministic Visual feature extractor denoted by $\Phi_v$. To enhance conciseness in written expression, we will use $\Phi_v$ to represent $\Phi_v(X)$ in this proof without further elaboration. Thus, we can write that

$$\begin{aligned} \left| \hat{I}_{\mathcal{S}}(Y; \Phi_v(X)) - I_{\mathcal{T}}(Y; \Phi_v(X)) \right| &\triangleq \left| \hat{I}_{\mathcal{S}}(Y; \Phi_v) - I_{\mathcal{T}}(Y; \Phi_v) \right| \\ &= \left| \hat{I}_{\mathcal{S}}(Y; \Phi_v) - I_{\mathcal{S}}(Y; \Phi_v) + I_{\mathcal{S}}(Y; \Phi_v) - I_{\mathcal{T}}(Y; \Phi_v) \right| \\ &\leq \underbrace{\left| \hat{I}_{\mathcal{S}}(Y; \Phi_v) - I_{\mathcal{S}}(Y; \Phi_v) \right|}_{\mathcal{A}_1} + \underbrace{\left| I_{\mathcal{S}}(Y; \Phi_v) - I_{\mathcal{T}}(Y; \Phi_v) \right|}_{\mathcal{A}_2} \end{aligned} \tag{13}$$

We know that the mutual information $I(Y; \Phi)$ is defined by:

$$I(Y; \Phi) \triangleq H(\Phi) - H(\Phi \mid Y) \tag{14}$$

where $H(\cdot)$ represents the Shannon information entropy. We firstly deal with the first term in the above inequality:

$$\begin{aligned} \mathcal{A}_1 &= \left| \hat{H}_{\mathcal{S}}(\Phi_v) - H_{\mathcal{S}}(\Phi_v) + H_{\mathcal{S}}(\Phi_v \mid Y) - \hat{H}_{\mathcal{S}}(\Phi_v \mid Y) \right| \\ &\leq \left| H_{\mathcal{S}}(\Phi_v \mid Y) - \hat{H}_{\mathcal{S}}(\Phi_v \mid Y) \right| + \left| \hat{H}_{\mathcal{S}}(\Phi_v) - H_{\mathcal{S}}(\Phi_v) \right| \end{aligned} \tag{15}$$

For the first term on the right side of Eq. 15, we can write that

$$
\begin{aligned}
& |H_{\mathcal{S}}(\Phi_v \mid Y) - \hat{H}_{\mathcal{S}}(\Phi_v \mid Y)| \\
&= \Big| \sum_y \big( p_{\mathcal{S}}(y) H_{\mathcal{S}}(\Phi_v \mid y) - \hat{p}_{\mathcal{S}}(y) \hat{H}_{\mathcal{S}}(\Phi_v \mid y) \big) \Big| \\
&= \Big| \sum_y \big( p_{\mathcal{S}}(y) H_{\mathcal{S}}(\Phi_v \mid y) - p_{\mathcal{S}}(y) \hat{H}_{\mathcal{S}}(\Phi_v \mid y) + p_{\mathcal{S}}(y) \hat{H}_{\mathcal{S}}(\Phi_v \mid y) - \hat{p}_{\mathcal{S}}(y) \hat{H}_{\mathcal{S}}(\Phi_v \mid y) \big) \Big| \\
&\leq \Big| \sum_y p_{\mathcal{S}}(y) \big( H_{\mathcal{S}}(\Phi_v \mid y) - \hat{H}_{\mathcal{S}}(\Phi_v \mid y) \big) \Big| + \Big| \sum_y \big( p_{\mathcal{S}}(y) - \hat{p}_{\mathcal{S}}(y) \big) \hat{H}_{\mathcal{S}}(\Phi_v \mid y) \Big|
\end{aligned}
$$

The first term on the right side of the above inequality can be bounded by

$$
\begin{aligned}
& \Big| \sum_y p_{\mathcal{S}}(y) \big( H_{\mathcal{S}}(\Phi_v \mid y) - \hat{H}_{\mathcal{S}}(\Phi_v \mid y) \big) \Big| \\
&\leq \Big| \sum_y p_{\mathcal{S}}(y) \sum_{\phi_v} \big( p_{\mathcal{S}}(\phi_v|y) \log(p_{\mathcal{S}}(\phi_v|y)) - \hat{p}_{\mathcal{S}}(\phi_v|y) \log(\hat{p}_{\mathcal{S}}(\phi_v|y)) \big) \Big| \\
&\leq \sum_y p_{\mathcal{S}}(y) \sum_{\phi_v} \xi \big( |p_{\mathcal{S}}(\phi_v|y) - \hat{p}_{\mathcal{S}}(\phi_v|y)| \big) \\
&= \sum_y p_{\mathcal{S}}(y) \sum_{\phi_v} \xi \Big( \Big| \sum_x p_{\mathcal{S}}(\phi_v|x) \big( p_{\mathcal{S}}(x|y) - \hat{p}_{\mathcal{S}}(x|y) \big) \Big| \Big) \\
&= \sum_y p_{\mathcal{S}}(y) \sum_{\phi_v} \xi \Big( \Big| \sum_x \big( p_{\mathcal{S}}(\phi_v|x) - A \big) \big( p_{\mathcal{S}}(x|y) - \hat{p}_{\mathcal{S}}(x|y) \big) \Big| \Big) \\
&\leq \sum_y p_{\mathcal{S}}(y) \sum_{\phi_v} \xi \Big( \big\| p_{\mathcal{S}}(X|y) - \hat{p}_{\mathcal{S}}(X|y) \big\| \big\| p_{\mathcal{S}}(\phi_v|X) - A \big\| \Big)
\end{aligned}
$$

where $A$ can be any constant. When we set $A \triangleq \frac{1}{|X|} \sum_x p_{\mathcal{S}}(\phi_v|x)$, we can get

$$
\Big| \sum_y p_{\mathcal{S}}(y) \big( H_{\mathcal{S}}(\Phi_v \mid y) - \hat{H}_{\mathcal{S}}(\Phi_v \mid y) \big) \Big| \leq \sum_y p_{\mathcal{S}}(y) \sum_{\phi_v} \xi \Big( \big\| p_{\mathcal{S}}(X|y) - \hat{p}_{\mathcal{S}}(X|y) \big\| \cdot \sqrt{V(p_{\mathcal{S}}(\phi_v|X))} \Big) \tag{16}
$$

where $\frac{1}{|X|} V(p_{\mathcal{S}}(\phi_v|X))$ describes the variance of the vector $p_{\mathcal{S}}(\phi_v|X)$. It is known that $\hat{H}_{\mathcal{S}}(\Phi_v) \geq \hat{H}_{\mathcal{S}}(\Phi_v \mid y)$ for any $y$, since conditioning cannot increase entropy Shamir et al. (2010). Therefore,

$$
\begin{aligned}
\Big| \sum_y \big( p_{\mathcal{S}}(y) - \hat{p}_{\mathcal{S}}(y) \big) \hat{H}_{\mathcal{S}}(\Phi_v \mid y) \Big| &\leq \big\| p_{\mathcal{S}}(Y) - \hat{p}_{\mathcal{S}}(Y) \big\| \Big| \sum_y \hat{H}_{\mathcal{S}}(\Phi_v) \Big| \\
&= \big\| p_{\mathcal{S}}(Y) - \hat{p}_{\mathcal{S}}(Y) \big\| \big( |Y| \hat{H}_{\mathcal{S}}(\Phi_v) \big)
\end{aligned} \tag{17}
$$

Because $\Phi_v(X) \in \mathcal{Z}$, we ca get that $\hat{H}_{\mathcal{S}}(\Phi_v) \leq \log(|\mathcal{Z}|)$ according to the definition of Shannon Information Entropy. Combining Eq. (16) and Eq. (17), we can get

$$
\begin{aligned}
H_{\mathcal{S}}(\Phi_v \mid Y) - \hat{H}_{\mathcal{S}}(\Phi_v \mid Y)| &\leq \sum_y p_{\mathcal{S}}(y) \sum_{\phi_v} \xi \Big( \big\| p_{\mathcal{S}}(X|y) - \hat{p}_{\mathcal{S}}(X|y) \big\| \cdot \sqrt{V(p_{\mathcal{S}}(\phi_v|X))} \Big) \\
&\quad + \big( |Y| \cdot \log(|\mathcal{Z}|) \big) \cdot \big\| p_{\mathcal{S}}(Y) - \hat{p}_{\mathcal{S}}(Y) \big\|
\end{aligned} \tag{18}
$$

On the other hand, we have

$$
\begin{aligned}
\left|H_{\mathcal{S}}(\Phi_v) - \hat{H}_{\mathcal{S}}(\Phi_v)\right| &= \left|\sum_{\phi_v}\big(p_{\mathcal{S}}(\phi_v)\log(p_{\mathcal{S}}(\phi_v)) - \hat{p}_{\mathcal{S}}(\phi_v)\log(\hat{p}_{\mathcal{S}}(\phi_v))\big)\right| \\
&\leq \sum_{\phi_v}\xi\big(\left|p_{\mathcal{S}}(\phi_v) - \hat{p}_{\mathcal{S}}(\phi_v)\right|\big) \\
&= \sum_{\phi_v}\xi\Big(\Big|\sum_x p_{\mathcal{S}}(\phi_v|x)\big(p_{\mathcal{S}}(x) - \hat{p}_{\mathcal{S}}(x)\big)\Big|\Big) \\
&= \sum_{\phi_v}\xi\Big(\Big|\sum_x \big(p_{\mathcal{S}}(\phi_v|x) - A\big)\big(p_{\mathcal{S}}(x) - \hat{p}_{\mathcal{S}}(x)\big)\Big|\Big) \\
&\leq \sum_{\phi_v}\xi\Big(\left\|p_{\mathcal{S}}(X) - \hat{p}_{\mathcal{S}}(X)\right\| \cdot \sqrt{V(p_{\mathcal{S}}(\phi_v|X))}\Big)
\end{aligned}
\tag{19}
$$

where the constant $A$ is chosen as $A \triangleq \frac{1}{|X|}\sum_x p_{\mathcal{S}}(\phi_v|x)$. Plugging Eq. (18) and Eq. (19) into Eq. (15), we can get

$$
\begin{aligned}
\mathcal{A}_1 &\leq \sum_y p_{\mathcal{S}}(y)\sum_{\phi_v}\xi\Big(\left\|p_{\mathcal{S}}(X|y) - \hat{p}_{\mathcal{S}}(X|y)\right\| \cdot \sqrt{V(p_{\mathcal{S}}(\phi_v|X))}\Big) \\
&\quad + \big(|Y|\log(|\mathcal{Z}|)\big)\cdot\left\|p_{\mathcal{S}}(Y) - \hat{p}_{\mathcal{S}}(Y)\right\| + \sum_{\phi_v}\xi\Big(\left\|p_{\mathcal{S}}(X) - \hat{p}_{\mathcal{S}}(X)\right\| \cdot \sqrt{V(p_{\mathcal{S}}(\phi_v|X))}\Big)
\end{aligned}
\tag{20}
$$

Subsequently, we can apply the concentration bound given in Proposition B.1 to $\left\|p_{\mathcal{S}}(X|y) - \hat{y}_{\mathcal{S}}(X|y)\right\|$, $\left\|p_{\mathcal{S}}(X) - \hat{p}_{\mathcal{S}}(X)\right\|$ and $\left\|p_{\mathcal{S}}(Y) - \hat{p}_{\mathcal{S}}(Y)\right\|$ for any $y$ in Eq. (20). To make sure the bounds hold simultaneously over these $|Y| + 2$ quantities, we replace $\delta$ in Eq. (11) by $\delta/(|Y| + 2)$ as in the proof of Theorem 3 in Shamir et al. (2010). Hence, with a probability at least $1 - \delta$ we have

$$
\begin{aligned}
\mathcal{A}_1 &\leq 2\sum_{\phi_v}\xi\left(\Big(2 + \sqrt{2\log((|Y| + 2)/\delta)}\Big)\sqrt{\frac{V\big(p_{\mathcal{S}}(\phi_v|X)\big)}{m}}\right) \\
&\quad + \frac{2 + \sqrt{2\log\big((|Y| + 2)/\delta\big)}}{\sqrt{m}}\cdot\big(|Y|\log(|\mathcal{Z}|)\big)
\end{aligned}
\tag{21}
$$

There exists a small constant $C$ that makes the following inequality hold:

$$
2 + \sqrt{2\log((|Y| + 2)/\delta)} \leq \sqrt{C\log(|Y|/\delta)}
$$

In addition, we know that the variance of any random variable that takes value in the range $[0, 1]$ is at most $\frac{1}{4}$. Since $\frac{1}{|X|}\sum_x V\big(p_{\mathcal{S}}(\phi_v|X)\big)$ is the variance of the distribution vector $p_{\mathcal{S}}(\phi_v|X)$, we have that $V\big(p_{\mathcal{S}}(\phi_v|X)\big) \leq |\mathcal{X}|/4, \forall\phi_v$.

Suppose that the size of training dataset (i.e., $m = |D_u|$) satisfying that

$$
m \geq \frac{C}{4}\log(|Y|/\delta)|X|e^2
\tag{22}
$$

Then, we can get

$$
\sqrt{\frac{C\log(|Y|/\delta)V(p_{\mathcal{S}}(\phi_v|X))}{m}} \leq \sqrt{\frac{C\log(|Y|/\delta)|X|}{4m}} \leq \frac{1}{e}, \forall\phi_v.
$$

We define that $\mathcal{V}(\phi_v) \triangleq C\log(|Y|/\delta)V\big(p_{\mathcal{S}}(\phi_v|X)\big)$, then we have that

$$
\begin{aligned}
\sum_{\phi_v}\xi\Big(\sqrt{\frac{\mathcal{V}(\phi_v)}{m}}\Big) &= \sum_{\phi_v}\sqrt{\frac{\mathcal{V}(\phi_v)}{m}}\log\Big(\sqrt{\frac{\mathcal{V}(\phi_v)}{m}}\Big) \\
&= \sum_{\phi_v}\sqrt{\frac{\mathcal{V}(\phi_v)}{m}}\log(\sqrt{m}) + \sqrt{\frac{1}{m}}\sqrt{\mathcal{V}(\phi_v)}\log\Big(\frac{1}{\sqrt{\mathcal{V}(\phi_v)}}\Big) \\
&\leq \sum_{\phi_v}\left(\sqrt{\frac{\mathcal{V}(\phi_v)}{m}}\log(\sqrt{m}) + \frac{1}{\sqrt{m}e}\right)
\end{aligned}
$$

Using the results proved in the proof of Theorem 3 in Shamir et al. (2010), we can have that $\sum_{\phi_v} \sqrt{\mathcal{V}(\phi_v)} \leq \sqrt{|\mathcal{X}||\Phi_v|}$. Therefore, we can write that

$$\sum_{\phi_v} \xi\left(\sqrt{\frac{C\log(|Y|/\delta)V(p_{\mathcal{S}}(\phi_v|X))}{m}}\right) \leq \frac{\sqrt{C\log(|Y|/\delta)|X||\Phi_v|}\log(m) + \frac{2}{e}|\Phi_v|}{2\sqrt{m}} \tag{23}$$

where $|\Phi_v|$ denote the size of the feature space from which $\Phi_v$ takes value. Recalling that $\Phi_v$ is used to represent $\Phi_v(X)$ where $\Phi_v$ itself is a deterministic feature extractor, we can conclude that $|\Phi_v| \leq |X|$. Thus, we can get

$$\begin{aligned}
\mathcal{A}_1 &\leq \frac{\sqrt{C\log(|Y|/\delta)|X|}\log(m) + \frac{2}{e}|X|}{\sqrt{m}} + \frac{\sqrt{C\log(|Y|/\delta)|Y|}\log(|\mathcal{Z}|)}{\sqrt{m}} \\
&= \frac{\sqrt{C\log(|Y|/\delta)}\Big(|X|\log(m) + |Y|\log(|\mathcal{Z}|)\Big) + \frac{2}{e}|X|}{\sqrt{m}}
\end{aligned} \tag{24}$$

As regard to the second term in Eq. (13), we can write that

$$\begin{aligned}
\mathcal{A}_2 &= |I_{\mathcal{T}}(Y; \Phi_v) - I_{\mathcal{S}}(Y; \Phi_v)| \\
&= \left|\sum_y \sum_{\phi_v} p_{\mathcal{T}}(y, \phi_v)\log\left(\frac{p_{\mathcal{T}}(y, \phi_v)}{p_{\mathcal{T}}(y)p_{\mathcal{T}}(\phi_v)}\right) - p_{\mathcal{S}}(y, \phi_v)\log\left(\frac{p_{\mathcal{S}}(y, \phi_v)}{p_{\mathcal{S}}(y)p_{\mathcal{S}}(\phi_v)}\right)\right| \\
&= \left|\sum_y \sum_{\phi_v} \Big(p_{\mathcal{T}}(y, \phi_v)\log\big(p_{\mathcal{T}}(y|\phi_v)\big) - p_{\mathcal{S}}(y, \phi_v)\log\big(p_{\mathcal{S}}(y|\phi_v)\big)\Big) + H_{\mathcal{T}}(Y) - H_{\mathcal{S}}(Y)\right|
\end{aligned} \tag{25}$$

As is commonly stated in the machine learning literature, the target variable $Y$ is an exogenous variable, which indicates that $p_{\mathcal{S}}(Y) = p_{\mathcal{T}}(Y)$. Therefore, we have that $\big|H_{\mathcal{S}}(Y) - H_{\mathcal{T}}(Y)\big| = 0$. In this way, we can write that

$$\begin{aligned}
\mathcal{A}_2 &\leq \left|\sum_y \sum_{\phi_v} \Big(p_{\mathcal{T}}(y, \phi_v)\log\big(p_{\mathcal{T}}(y|\phi_v)\big) - p_{\mathcal{S}}(y, \phi_v)\log\big(p_{\mathcal{S}}(y|\phi_v)\big)\Big)\right| \\
&= \left|\sum_y \sum_{\phi_v} \Big(p_{\mathcal{T}}(y, \phi_v)\log\big(p_{\mathcal{T}}(y|\phi_v)\big) - p_{\mathcal{T}}(y, \phi_v)\log\big(p_{\mathcal{S}}(y|\phi_v)\big) + p_{\mathcal{T}}(y, \phi_v)\log\big(p_{\mathcal{S}}(y|\phi_v)\big) - p_{\mathcal{S}}(y, \phi_v)\log\big(p_{\mathcal{S}}(y|\phi_v)\big)\Big)\right| \\
&\leq \left|\sum_y \sum_{\phi_v} p_{\mathcal{T}}(y, \phi_v)\log\left(\frac{p_{\mathcal{T}}(y|\phi_v)}{p_{\mathcal{S}}(y|\phi_v)}\right)\right| + \left|\sum_y \sum_{\phi_v} \big(p_{\mathcal{T}}(y, \phi_v) - p_{\mathcal{S}}(y, \phi_v)\big)\log\big(p_{\mathcal{S}}(y|\phi_v)\big)\right| \\
&= \mathcal{KL}\big(p_{\mathcal{T}}(Y \mid \Phi_v)\|p_{\mathcal{S}}(Y \mid \Phi_v)\big) + \underbrace{\left|\sum_y \sum_{\phi_v} \big(p_{\mathcal{T}}(y, \phi_v) - p_{\mathcal{S}}(y, \phi_v)\big)\log\big(p_{\mathcal{S}}(y|\phi_v)\big)\right|}_{\mathcal{B}}
\end{aligned}$$

According to the above equation, we have that

$$\mathcal{B}^2 = \left\|\sum_y \sum_{\phi_v} \big(p_{\mathcal{T}}(y, \phi_v) - p_{\mathcal{S}}(y, \phi_v)\big)\log\big(p_{\mathcal{S}}(y|\phi_v)\big)\right\|^2$$

Using the Jensen's inequality, we can get

$$\begin{aligned}
\mathcal{B}^2 &\leq |Y|\sum_y \left\|\sum_{\phi_v} \big(p_{\mathcal{T}}(y, \phi_v) - p_{\mathcal{S}}(y, \phi_v)\big)\log\big(p_{\mathcal{S}}(y|\phi_v)\big)\right\|^2 \\
&\leq |Y|\sum_y \sum_{\phi_v} p(\phi_v)\left\|\big(p_{\mathcal{T}}(y|\phi_v) - p_{\mathcal{S}}(y|\phi_v)\big)\log\big(p_{\mathcal{S}}(y|\phi_v)\big)\right\|^2, \\
&\leq |Y|C_{\mathcal{S}}^2\sum_y \sum_{\phi_v} p(\phi_v)\big\|p_{\mathcal{T}}(y|\phi_v) - p_{\mathcal{S}}(y|\phi_v)\big\|^2
\end{aligned}$$

where $C_S$ denotes a constant satisfying that $C_S = \max_{(\phi_v, y) \in (\Phi_v, Y)} \big| \log\big(p_S(y|\phi_v)\big) \big|$. We know that $\log(\cdot)$ is a concave function, therefore we can get

$$
\begin{aligned}
\mathcal{B}^2 &\leq |Y| C_S^2 \sum_y \sum_{\phi_v} p(\phi_v) \big\| p_\mathcal{T}(y|\phi_v) - p_S(y|\phi_v) \big\| \big\| \log\big(p_\mathcal{T}(y|\phi_v)\big) - \log\big(p_S(y|\phi_v)\big) \big\| \\
&= |Y| C_S^2 \sum_y \sum_{\phi_v} p(\phi_v) \big(p_\mathcal{T}(y|\phi_v) - p_S(y|\phi_v)\big) \Big( \log\big(p_\mathcal{T}(y|\phi_v)\big) - \log\big(p_S(y|\phi_v)\big) \Big) \\
&= |Y| C_S^2 \sum_y \sum_{\phi_v} p(\phi_v) \bigg( p_\mathcal{T}(y|\phi_v) \log\Big(\frac{p_\mathcal{T}(y|\phi_v)}{p_S(y|\phi_v)}\Big) - p_S(y|\phi_v) \log\Big(\frac{p_\mathcal{T}(y|\phi_v)}{p_S(y|\phi_v)}\Big) \bigg) \\
&= |Y| C_S^2 \Big( \mathcal{KL}\big(p_\mathcal{T}(Y \mid \Phi_v) \| p_S(Y \mid \Phi_v)\big) + \mathcal{KL}\big(p_S(Y \mid \Phi_v) \| p_\mathcal{T}(Y \mid \Phi_v)\big) \Big).
\end{aligned}
$$

Consequently, we can get that

$$
\begin{aligned}
\mathcal{A}_2 &\leq \mathcal{KL}\big(p_\mathcal{T}(Y \mid \Phi_v) \big\| p_S(Y \mid \Phi_v)\big) \\
&\quad + \sqrt{|Y| C_S^2 \Big( \mathcal{KL}\big(p_\mathcal{T}(Y \mid \Phi_v) \| p_S(Y \mid \Phi_v)\big) + \mathcal{KL}\big(p_S(Y \mid \Phi_v) \| p_\mathcal{T}(Y \mid \Phi_v)\big) \Big)} \qquad (26) \\
&\leq \mathcal{J}\big(p_\mathcal{T}(Y \mid \Phi_v), p_S(Y \mid \Phi_v)\big) + \sqrt{|Y| C_S^2 \mathcal{J}\big(p_\mathcal{T}(Y \mid \Phi_v), p_S(Y \mid \Phi_v)\big)}
\end{aligned}
$$

where $\mathcal{J}(p, q)$ denotes the Jeffrey's divergence between probability $p$ and $q$ which is defined by

$$
\mathcal{J}\big(p_\mathcal{T}(Y \mid \Phi_v), p_S(Y \mid \Phi_v)\big) \triangleq \mathcal{KL}\big(p_\mathcal{T}(Y \mid \Phi_v) \| p_S(Y \mid \Phi_v)\big) + \mathcal{KL}\big(p_S(Y \mid \Phi_v) \| p_\mathcal{T}(Y \mid \Phi_v)\big)
$$

With Equation (24) and Equation (26), we can conclude that

$$
\begin{aligned}
|\hat{I}_S(Y; \Phi_v(X)) - I_\mathcal{T}(Y; \Phi_v(X))| &\leq \frac{\sqrt{C \log(|Y|/\delta)} \Big( |X| \log(m) + |Y| \log(|\mathcal{Z}|) \Big) + \frac{2}{e} |X|}{\sqrt{m}} \\
&\quad + \mathcal{J}\big(p_\mathcal{T}(Y \mid \Phi_v), p_S(Y \mid \Phi_v)\big) + \sqrt{|Y| C_S^2 \mathcal{J}\big(p_\mathcal{T}(Y \mid \Phi_v), p_S(Y \mid \Phi_v)\big)}
\end{aligned}
$$
$$(27)$$

When Assumption 4.1 is satisfied, we have that $Y \perp\!\!\!\perp \Phi_v \mid \Phi_t$. Thus, we can get that $p_S(Y \mid \Phi_v, \Phi_t) = p_S(Y \mid \Phi_t), \forall \Phi_t, \Phi_v$ and $p_\mathcal{T}(Y \mid \Phi_v, \Phi_t) = p_\mathcal{T}(Y \mid \Phi_t), \forall \Phi_t, \Phi_v$. In other words, we can derive that

$$
p_S(Y, \Phi_v) = \sum_{\phi_t} p_S(Y, \phi_t, \Phi_v) = \sum_{\phi_t} p_S(Y \mid \phi_t, \Phi_v) p_S(\phi_t, \Phi_v) = \sum_{\phi_t} p_S(Y \mid \phi_t) p_S(\phi_t, \Phi_v).
$$

That is, $p_S(Y \mid \Phi_v) = \sum_{\phi_t} p_S(Y \mid \phi_t) p_S(\phi_t \mid \Phi_v)$. Similarly, the probability distribution $p_\mathcal{T}(Y \mid \Phi_v)$ can be rewrite as $p_\mathcal{T}(Y \mid \Phi_v) = \sum_{\phi_t} p_\mathcal{T}(Y \mid \phi_t) p_\mathcal{T}(\phi_t \mid \Phi_v)$. Plugging these two equations into $\mathcal{KL}\big(p_S(Y \mid \Phi_v) \| p_\mathcal{T}(Y \mid \Phi_v)\big)$, we can obtain that

$$
\begin{aligned}
\mathcal{KL}\big(p_S(Y \mid \Phi_v) \| p_\mathcal{T}(Y \mid \Phi_v)\big) &= \sum_y \sum_{\phi_v} p_S(y, \phi_v) \log\Big(\frac{p_S(y \mid \phi_v)}{p_\mathcal{T}(y \mid \phi_v)}\Big) \\
&= \sum_y \sum_{\phi_v} p(\phi_v) \sum_{\phi_t} p_S(y \mid \phi_t) p_S(\phi_t \mid \phi_v) \log\Big(\frac{\sum_{\phi_t} p_S(y \mid \phi_t) p_S(\phi_t \mid \phi_v)}{\sum_{\phi_t} p_\mathcal{T}(y \mid \phi_t) p_\mathcal{T}(\phi_t \mid \phi_v)}\Big).
\end{aligned}
$$

Here we consider a real-valued function $\zeta(x) = x \log(x)$ which is a convex function. Then, we can write that

$$
\begin{aligned}
\mathcal{KL}\big(p_S(Y \mid \Phi_v) \| p_\mathcal{T}(Y \mid \Phi_v)\big) &= \sum_y \sum_{\phi_v} p(\phi_v) \sum_{\phi_t} p_\mathcal{T}(y \mid \phi_t) p_\mathcal{T}(\phi_t \mid \phi_v) \zeta\Big(\frac{\sum_{\phi_t} p_S(y \mid \phi_t) p_S(\phi_t \mid \phi_v)}{\sum_{\phi_t} p_\mathcal{T}(y \mid \phi_t) p_\mathcal{T}(\phi_t \mid \phi_v)}\Big) \\
&= \sum_y \sum_{\phi_v} p(\phi_v) \sum_{\phi_t} p_\mathcal{T}(y \mid \phi_t) p_\mathcal{T}(\phi_t \mid \phi_v) \zeta\Big( \sum_{\phi_t} \frac{p_\mathcal{T}(y \mid \phi_t) p_\mathcal{T}(\phi_t \mid \phi_v)}{\sum_{\phi_t} p_\mathcal{T}(y \mid \phi_t) p_\mathcal{T}(\phi_t \mid \phi_v)} \cdot \frac{p_S(y \mid \phi_t) p_S(\phi_t \mid \phi_v)}{p_\mathcal{T}(y \mid \phi_t) p_\mathcal{T}(\phi_t \mid \phi_v)}\Big) \\
&\leq \sum_y \sum_{\phi_v} p(\phi_v) \sum_{\phi_t} p_\mathcal{T}(y \mid \phi_t) p_\mathcal{T}(\phi_t \mid \phi_v) \sum_{\phi_t} \frac{p_\mathcal{T}(y \mid \phi_t) p_\mathcal{T}(\phi_t \mid \phi_v)}{\sum_{\phi_t} p_\mathcal{T}(y \mid \phi_t) p_\mathcal{T}(\phi_t \mid \phi_v)} \zeta\Big(\frac{p_S(y \mid \phi_t) p_S(\phi_t \mid \phi_v)}{p_\mathcal{T}(y \mid \phi_t) p_\mathcal{T}(\phi_t \mid \phi_v)}\Big) \\
&= \sum_y \sum_{\phi_v} p(\phi_v) \sum_{\phi_t} p_\mathcal{T}(y \mid \phi_t) p_\mathcal{T}(\phi_t \mid \phi_v) \zeta\Big(\frac{p_S(y \mid \phi_t) p_S(\phi_t \mid \phi_v)}{p_\mathcal{T}(y \mid \phi_t) p_\mathcal{T}(\phi_t \mid \phi_v)}\Big)
\end{aligned}
$$

According to the definition of $\zeta(x) = x \log(x)$, we can get that

$$\mathcal{KL}\big(p_{\mathcal{S}}(Y \mid \Phi_v) \| p_{\mathcal{T}}(Y \mid \Phi_v)\big)$$

$$\leq \sum_y \sum_{\phi_v} p(\phi_v) \sum_{\phi_t} p_{\mathcal{S}}(y \mid \phi_t) p_{\mathcal{S}}(\phi_t \mid \phi_v) \log\left(\frac{p_{\mathcal{S}}(y \mid \phi_t) p_{\mathcal{S}}(\phi_t \mid \phi_v)}{p_{\mathcal{T}}(y \mid \phi_t) p_{\mathcal{T}}(\phi_t \mid \phi_v)}\right)$$

$$= \sum_y \sum_{\phi_v} p(\phi_v) \sum_{\phi_t} p_{\mathcal{S}}(y \mid \phi_t) p_{\mathcal{S}}(\phi_t \mid \phi_v) \log\left(\frac{p_{\mathcal{S}}(y \mid \phi_t)}{p_{\mathcal{T}}(y \mid \phi_t)}\right)$$

$$+ \sum_y \sum_{\phi_v} p(\phi_v) \sum_{\phi_t} p_{\mathcal{S}}(y \mid \phi_t) p_{\mathcal{S}}(\phi_t \mid \phi_v) \log\left(\frac{p_{\mathcal{S}}(\phi_t \mid \phi_v)}{p_{\mathcal{T}}(\phi_t \mid \phi_v)}\right)$$

$$= \sum_y \sum_{\phi_v} \sum_{\phi_t} p_{\mathcal{S}}(y, \phi_t, \phi_v) \log\left(\frac{p_{\mathcal{S}}(y \mid \phi_t)}{p_{\mathcal{T}}(y \mid \phi_t)}\right)$$

$$+ \sum_y \sum_{\phi_v} \sum_{\phi_t} p_{\mathcal{S}}(y, \phi_t, \phi_v) \log\left(\frac{p_{\mathcal{S}}(\phi_t \mid \phi_v)}{p_{\mathcal{T}}(\phi_t \mid \phi_v)}\right)$$

$$= \sum_y \sum_{\phi_t} p_{\mathcal{S}}(y, \phi_t) \log\left(\frac{p_{\mathcal{S}}(y \mid \phi_t)}{p_{\mathcal{T}}(y \mid \phi_t)}\right) + \sum_{\phi_t} \sum_{\phi_v} p_{\mathcal{S}}(\phi_t, \phi_v) \log\left(\frac{p_{\mathcal{S}}(\phi_t \mid \phi_v)}{p_{\mathcal{T}}(\phi_t \mid \phi_v)}\right)$$

$$= \mathcal{KL}\big(p_{\mathcal{S}}(Y \mid \Phi_t) \| p_{\mathcal{T}}(Y \mid \Phi_t)\big) + \mathcal{KL}\big(p_{\mathcal{S}}(\Phi_t \mid \Phi_v) \| p_{\mathcal{T}}(\Phi_t \mid \Phi_v)\big)$$

Similarly, we can also derive that

$$\mathcal{KL}\big(p_{\mathcal{T}}(Y \mid \Phi_v) \| p_{\mathcal{S}}(Y \mid \Phi_v)\big) \leq \mathcal{KL}\big(p_{\mathcal{T}}(Y \mid \Phi_t) \| p_{\mathcal{S}}(Y \mid \Phi_t)\big) + \mathcal{KL}\big(p_{\mathcal{T}}(\Phi_t \mid \Phi_v) \| p_{\mathcal{S}}(\Phi_t \mid \Phi_v)\big)$$

Therefore, we conclude that

$$\mathcal{J}\big(p_{\mathcal{T}}(Y \mid \Phi_v) \| p_{\mathcal{S}}(Y \mid \Phi_v)\big) \leq \mathcal{J}\big(p_{\mathcal{T}}(Y \mid \Phi_t) \| p_{\mathcal{S}}(Y \mid \Phi_t)\big) + \mathcal{J}\big(p_{\mathcal{T}}(\Phi_t \mid \Phi_v) \| p_{\mathcal{S}}(\Phi_t \mid \Phi_v)\big).$$

Plugging this inequality into inequality 27, we can finally get

$$\big|I_{\mathcal{T}}(Y; \Phi_v(X)) - \hat{I}_{\mathcal{S}}(Y; \Phi_v(X))\big| \leq \frac{\sqrt{C \log(|\mathcal{Y}|/\delta)}\Big(|\mathcal{X}| \log(m) + |\mathcal{Y}| \log(|\mathcal{Z}|)\Big) + \frac{2}{e}|\mathcal{X}|}{\sqrt{m}}$$

$$+ \quad \mathcal{J}(Y | \Phi_t) + \sqrt{C|\mathcal{Y}| \mathcal{J}(Y | \Phi_t)} + \mathcal{J}(\Phi_t | \Phi_v) + \sqrt{C|\mathcal{Y}| \mathcal{J}(\Phi_t | \Phi_v)},$$

Thus, we complete the proof of Theorem 4.2. $\qquad\square$

## C   MORE DETAILS ABOUT PNS AND PNS MODELING

Probability of Necessity and Sufficiency (PNS) describe the probability with which a variable is the necessary and sufficient cause of another variable. The formal definition of PNS is given as follows.

**Definition C.1** (Probability of Necessity and Sufficiency (Pearl, 2009)). *Let the specific implementations of causal variable $\Phi$ as $\phi$ and $\bar{\phi}$, where $\phi \neq \bar{\phi}$. The probability with which variable $\Phi$ is the necessary and sufficient cause of variable $Y$ on test data distribution $P_{\mathcal{T}}$ is given by:*

$$PNS(Y, \Phi) := \underbrace{P_{\mathcal{T}}(Y_{do(\Phi=\phi)} = y \mid \Phi = \bar{\phi}, Y \neq y)}_{sufficiency} P_{\mathcal{T}}(\Phi = \bar{\phi}, Y \neq y)$$

$$+ \underbrace{P_{\mathcal{T}}(Y_{do(\Phi=\bar{\phi})} \neq y \mid \Phi = \phi, Y = y)}_{necessity} P_{\mathcal{T}}(\Phi = \phi, Y = y), \tag{28}$$

*where $do(\Phi = \phi)$ (do-operator) indicates that the manipulable variable $\Phi$ is forced to be a fixed value $\Phi = \phi$.*

Since the probability of necessity and sufficiency is defined based on counterfactual distributions, it is usually intractable to estimate the PNS of two variables. Therefore, we need some assumptions to facilitate the practical calculation of PNS.

**Assumption C.2** (Exogeneity (Pearl, 2009; Yang et al., 2023b)). *Variable $\Phi$ is exogenous relative to variable $Y$ with respect to the source domain $\mathcal{S}$ and target domain $\mathcal{T}$, if the intervention probability is identified by conditional probability, i.e., $P_{\mathcal{S}}(Y_{do(\Phi=\phi)} = y) = P_{\mathcal{S}}(Y = y \mid \Phi = \phi)$ and $P_{\mathcal{T}}(Y_{do(\Phi=\phi)} = y) = P_{\mathcal{T}}(Y = y \mid \Phi = \phi)$.*

**Assumption C.3** (Monotonicity (Pearl, 2009; Yang et al., 2023b)). *Variable $Y$ is monotonic relative to variable $\Phi$ if and only if either $P(Y_{do(\Phi=\phi)} = y, Y_{do(\Phi=\bar{\phi})} \neq y) = 0$ or $P(Y_{do(\Phi=\phi)} \neq y, Y_{do(\Phi=\bar{\phi})} = y) = 0$ holds.*

Exogeneity defined in Assumption C.2 bridges the gap between the intractable intervention probability and the computable conditional probability, while monotonicity defined in Assumption C.3 guarantees that the causal variable $\Phi$ has monotonic effect on variable $Y$. With these two assumptions, we can obtain a useful lemma as follows.

**Lemma C.4** (Pearl (2009); Yang et al. (2023b)). *If variable $\Phi$ is exogenous relative to variable $Y$, and $Y$ is monotonic relative to $\Phi$, we can get*

$$PNS(Y, \Phi) = \underbrace{P_{\mathcal{T}}(Y = y \mid \Phi = \phi)}_{sufficiency} - \underbrace{P_{\mathcal{T}}(Y = y \mid \Phi = \bar{\phi})}_{necessity}. \tag{29}$$

# D MORE EXPERIMENTAL RESULTS

**Implementation Details**  In all experiments, we use the publicly available CLIP model with the ResNet-50 (He et al., 2016) and ViT-B/32 (Dosovitskiy, 2020) as the backbone models. The prompt used in all methods has 8 learnable tokens and initialized as the default one "a photo of". When comparing the performance with baselines, we optimize the prompts for 50 epochs with SGD optimizer and a cosine decay learning rate scheduler, the initial learning rate is 0.002. The batch size of images is 32 on all datasets. For LogicAl-PT, unless otherwise specified, the value of hyper-parameters $\alpha$ and $\beta$ are 10.0 and 1.0 for CelebA; 20.0, 1.0 for ImageNet-1K; 3.0 and 2.0 for WaterBird.

**Computational Efficiency**  We analyze and compare the computational overhead of our method with several existing methods to verify the computational efficiency of the proposed LogicAl-PT. The results are presented in the following table.

Table 4: Evaluation results on computational overhead of our method LogicAl-PT and the state-of-the-art competitors. 'Params' denotes the number of learnable parameters while 'FLOPS' represents 'Floating Point Operations'.

| Method | Params | Params + %CLIP | FLOPS | FLOPS + %CoOp |
|--------|--------|----------------|-------|---------------|
| CoOp | 2048 | 0.004% | 354.50G | - |
| ERM | 0.514M | 1.05% | 354.53G | 0.01% |
| CoOPood | 1.026M | 2.10% | 354.56G | 0.02% |
| LogicAl-PT | 1.028M | 2.10% | 354.57G | 0.02% |

We can see the the overall parameters and Floating Point Operations (FLOPS) of our LogicAl-PT are only 2.1% and 0.02% higher than those of CLIP and CoOp, respectively. Compared with the improvement in out-of-distribution generalization performance, our method LogicAl-PT impressive computational efficiency in terms of the number of parameters and FLOPS.

**Adaptation from Causal Representation Learning**  To enhance the motivation for utilizing PNS modeling to improve out-of-distribution generalization during prompt tuning of VLMs, we adapt two representative causal representation learning methods from invariant learning: IRM (Arjovsky et al., 2019) and IB-IRM (Ahuja et al., 2021). They mitigate spurious correlations by ensuring the invariance of the conditional probability of the label $Y$ given the causal representation across varied training environments. The evaluation is conducted using ResNet-50 as the backbone model. The experimental results on four datasets are list as follows:

We can find that LogicAl-PT outperforms the typical causally invariant representation learning methods. The underlying reason stems from the advantage of 'sufficient and neccesary' causal representa-

Table 5: Performance comparison among our method LogicAl-PT and the prevalent schemes adapted from two single-modal causal presentation learning methods.

| Datasets | Waterbird | | CelebA | | ImageNet-1K | | PACS | |
|---|---|---|---|---|---|---|---|---|
| Test Acc (%) | Worst | Avg | Worst | Avg | Worst | Avg | Worst | Avg |
| ERM | 54.7 | 84.1 | 26.7 | 78.2 | 80.5 | 88.5 | 80.0 | 92.6 |
| IRM | 64.7 | 83.9 | 67.1 | 86.2 | 87.9 | 93.6 | 80.7 | **93.8** |
| IB-IRM | 65.3 | 84.3 | 67.9 | 85.8 | 88.3 | 93.9 | 81.2 | 93.4 |
| LogicAl-PT | **67.5** | **86.2** | **69.9** | **87.3** | **90.2** | **95.1** | **82.4** | 93.7 |

tion over traditional causal representation, which also forms the motivation for proposing LogicAl-PT for prompt tuning of VLMs. Prevalent causal representation learning methods primarily aim to mitigate non-causal spurious correlations. In contrast, the concept of 'sufficiency and necessity' goes further by excluding not only non-causal spurious correlations but also causal relationships that are 'sufficient but not necessary' or 'necessary but not sufficient'. We provide specific examples to clarify these types of relationships and explain why only 'sufficient and necessary' relations remain stable across diverse data distributions in Figure 4.

