# OpenReview forum: "Cross-modal Mitigation of Spurious Correlation for Prompt-tuning in VLMs with Causally Motivated Logic Alignment"
_ICLR.cc/2025/Conference — Submitted to ICLR 2025_

### Official Review · Reviewer_Rh7Y · 2024-10-30

**Soundness:** 2
**Presentation:** 2
**Contribution:** 2
**Rating:** 3
**Confidence:** 4

**Summary:**

This article introduces the concept of logical alignment to address the cross-modal mitigation problem of spurious correlation for prompt adjustment in visual language models. To achieve this, the authors maximize the probability of necessity and sufficiency corresponding to cross-modal and textual logical alignment. Theoretical analysis proves that the proposed method has a tighter generalization error bound compared to existing approaches. Performance is analyzed across a few different test data distributions, and components of the method are ablated.

**Strengths:**

The generalization error bond of the presented method is provided with theory analysis in Appendix A.

The novelty of the method is how to integrate the probability of necessity and sufficiency in multi-modal learning.

**Weaknesses:**

The paper is not easy to follow. This is due to the symbols being confused, e.g., $\Phi$ denotes the filter in Cross-modal logic alignment but visual representation space in Textual logic alignment.
While the first half of the paper explains the idea and motivation well, creating a rightful sense of expectation of the result, the section on the results somewhat comes short of delivering the findings with a bang.  After reading the first half I was excited to read the next pages to find "Where are those indeed integrated areas for boosting the expected performance ", and tingling with an expectation of learning something new. But then, for some reason, the Overall Performance and Ablation Study sections are very timid and just present dry numbers for each of the tests that were planned.

1. It would be helpful to include a better explanation of what the "spurious correlation in vision-language models" is exactly. Maybe a picture.
2.The paper borrowed too much content from the existing papers and it could be removed by referring these papers. Even so, the motivation is not clear since the authors employ PNS without explanation.

3. Page 6, Section 4.1: The NSC feature shown Figure 1 is not explained by the authors themselves how to make use and take advantage of this.

4. Compared to CoOPood,  it seems that the proposed method exploits PNS terms instead of mutual information to align cross-modal representations. I am wondering how effective the PNS term is in cross-modal mitigation of spurious correlation.
5. Why was it necessary to do textual logic alignment?

**Questions:**

I would like to hear the authors' discussion regarding the three weakness that I highlikeed above:

1. Usefulness of the PNS term.

2. Usefulness of the textual logic alignment.

3. How to extractive NSC features in textual and visual modality.

---

> ### Author Response · Authors · 2024-11-26
> **Response to Reviewer Rh7Y (Part 1)**
>
> Thank you very much for your valuable comments and feedback, which significantly contributes to improving the quality of this paper. Detailed responses to your concerns and questions are listed below.
>
> >**W1: Confused symbols: $\Phi$ denotes the filter in Cross-modal logic alignment but visual representation space in Textual logic alignment.**
>
> **Answer:** Thanks for your suggestions. To distinguish the filter from the representation space, we use a different symbol ($\mathbf{h}$) to denote the filter in the revised paper. Meanwhile, $\Phi_t$ and $\Phi_v$ continue to represent the textual and visual representation spaces, respectively. Accordingly, the corresponding symbols in Figure 1 (on page 5) have also been updated.
>
> &nbsp;
>
> >**W2: Timid evaluation section.**
>
> **Answer:** To comprehensively validate the effectiveness of the proposed method, we have expanded the evaluation section (Section 5) by adding the following contents into the main text of the revised paper:
>
> **1)** Two state-of-the-art prompt tuning methods as **baselines** (PromptSRC [1] and DePT [2]), in Section 5.2 (on page 8);
>
> **2)** Experiments on another **backbone model (i.e., ViT-B/32)**, in Section 5.2 (on page 8);
>
> **3)** **Visualization experiments** to evaluate whether the proposed LogicAl-PT effectively mitigates cross-modal spurious correlations and enhances logical alignment between visual representation and text label, in Section 5.3 (on page 8-9);
>
> **4)** Visual explanation to illustrate **the necessity of the proposed textual logic alignment**, in Section 5.4 (on page 9);
>
> **5)** **Ablation study** on the sensitivity of hyper-parameters, in Section 5.4 (on page 10).
>
> [1] Khattak, Muhammad Uzair, et al. "Self-regulating prompts: Foundational model adaptation without forgetting." Proceedings of the IEEE/CVF International Conference on Computer Vision. 2023.
>
> [2] Zhang, Ji, et al. "Dept: Decoupled prompt tuning." Proceedings of the IEEE/CVF Conference on Computer Vision and Pattern Recognition. 2024.
>
> &nbsp;
>
> >**W3: The motivation is not clear since the authors employ PNS without explanation; It would be helpful to include a better explanation of what the "spurious correlation in vision-language models" is exactly.**
>
> **Answer:** To better **clarify the motivation** for using logic alignment to integrate cross-modal mitigation of spurious correlations and cross-modal feature alignment, we summarize four **specific examples in Figure 4 (on page 14 of Appendix A)** to address the following questions:
>
> **1)** What the "**spurious correlation** in vision-language models" is exactly.
>
> **2)** Apart from mitigation of spurious correlations, **why corss-modal logic alignment (i.e., sufficiency and necessity) is also necessary** for enhancing out-of-distribution generalization performance in vision-language models?
>
> &nbsp;
>
> >**W4: Too much content from the existing papers and it could be removed by referring these papers.**
>
> **Answer:** Very good suggestion. We have removed the redundant parts in Section 3.2 (PNS) and Section 3.3 (PNS modeling). Additional details about PNS have been moved to Appendix C for further reference.
>
> &nbsp;
>
> >**W5: Section 4.1: The NSC feature shown in Figure 1 is not explained by the authors themselves how to make use and take advantage of this.**
>
> **Answer:** Sorry, we missed providing the explanation for the NSC feature in Figure 1. "NSC" represents "necessary and sufficient cause''. Specifically, the NSC features in textual and visual modalities are given by $f([Q,CLASS])$ and $h(g(X))$, respectively. The interventions in textual and visual modalities are given by $f([\bar{Q},CLASS])$ and $\bar{h}(g(X))$, respectively.
>
> At the training stage, "NSC" features are optimized by adjusting the learnable prompt $Q$ and filter $h$ using the proposed objective (10), as stated on line 290, page 6.
>
> At the inference stage, predictions are made using the cosine similarity between textual and visual "NSC" features.
>
> **Advantage:** By optimizing the textual and visual "NSC" features through the proposed objective (10), the optimal textual "NSC" features become logically aligned with both text labels and the optimal visual "NSC" features. Leveraging these features for prediction effectively eliminates spurious correlations and improves out-of-distribution generalization performance.

---

> ### Author Response · Authors · 2024-11-26
> **Response to Reviewer Rh7Y (Part 2)**
>
> >**W6: Compared to CoOPood, it seems that the proposed method exploits PNS terms instead of mutual information to align cross-modal representations. I am wondering how effective the PNS term is in cross-modal mitigation of spurious correlation.**
>
> **Answer:**
>
> **1) Clarification on CoOPood:** In CoOPood, the conditional mutual information is specifically employed to disentangle visual invariant features from visual spurious features, rather than facilitating cross-modal feature alignment. In contrast, CoOPood employs the assumption that the spurious correlation between visual spurious features and text label follows approximately uniform probability distributions to achieve cross-modal alignment. Therefore, the mitigation of spurious correlations and cross-modal alignment in CoOPood cannot be ensured if the assumption is not met. In comparison, our method does not rely on any assumptions about spurious correlations.
>
> **2) Effectiveness/Usefulness of the PNS term:** To verify that the tuned models developed by our method LogicAl-PT exploit the necessary and sufficient features rather than spurious features, we sample some data instances to generate **visual explanations** for the selected model using Grad-CAM [3]. The commonly used Grad-CAM can produce a localization map which highlights the important regions in the input image that a deep learning model depends on for predicting the label.
>
> **The visualization results are displayed in Figure 2 (on page 9), and the detailed analysis on the visualization results is provided on line 410-431, page 8.** In summary, visualization results demonstrate the proposed LogicAl-PT can effectively exploit the 'sufficient and necessary' features and mitigate the unstable spurious features, including non-causal spurious features, 'sufficient but not necessary' features and 'necessary but not sufficient' features. This explains why LogicAl-PT achieves superior out-of-distribution generalization performance, delivering more consistent results across diverse data distributions compared to its competitors.
>
> [3] Selvaraju, Ramprasaath R., et al. "Grad-cam: Visual explanations from deep networks via gradient-based localization." Proceedings of the IEEE international conference on computer vision. 2017.
>
> &nbsp;
>
> >**W7: Usefulness of the textual logic alignment: Why was it necessary to do textual logic alignment?**
>
> **Answer:**
>
> **1) Qualitative Analysis:** Since the textual representations (corresponding to variable $\Phi_t$) are the class-wise mapping from the text labels, the sufficiency of variable $Y$ for variable $\Phi_t$ (i.e., $Y\Rightarrow \Phi_t$) is naturally guaranteed while the reverse $Y\Leftarrow \Phi_t$ is not ensured. In other words, textual representations ($\Phi_t$) must be necessary causes for variable $Y$, but they don't have to be sufficient causes for variable $Y$. Therefore, textual logic alignment is proposed to enhance the sufficiency of text representations ($\Phi_t$) for label $Y$. Accordingly, when cross-modal logic alignment (i.e., $\Phi_t \Leftrightarrow \Phi_v$) is achieved, combining textual logic alignment can mitigate the visual features that are not sufficient for variable $Y$.
>
> **2) Experimental Validation:** To investigate the actual role that textual logic alignment serves, we visualize the features which is utilized by the model tuned without textual logic alignment (w/o TLA), i.e., $\beta=0$. In particular, when we set $\beta=0$, $\alpha$ is tuned to its optimal value, i.e., the cross-modal logic alignment ($\Phi_t \Leftrightarrow \Phi_v$) is enhanced. **The visualization results are displayed in Figure 3 on page 10**. From the visualization results, we find that adding textual logic alignment mitigates the visual features which are not sufficient for predicting $Y$. **Therefore, the above qualitative analysis is validated by the visualization results.**
>
> Detailed visualization results and analysis are provided on line 483-513, page 10.

---

> > ### Author Response · Authors · 2024-11-30
> > **Response to Reviewer Rh7Y (Part 3)**
> >
> > >**Q1: Usefulness of the PNS term.**
> >
> > **Answer:**
> > To verify that the tuned models developed by our method LogicAl-PT exploit the necessary and sufficient features rather than spurious features, we sample some data instances to generate **visual explanations** for the selected model using Grad-CAM [3]. The commonly used Grad-CAM can produce a localization map which highlights the important regions in the input image that a deep learning model depends on for predicting the label.
> >
> > **The visualization results are displayed in Figure 2 (on page 9), and the detailed analysis on the visualization results is provided on line 410-431, page 8.** In summary, visualization results demonstrate the proposed LogicAl-PT can effectively exploit the 'sufficient and necessary' features and mitigate the unstable spurious features, including non-causal spurious features, 'sufficient but not necessary' features and 'necessary but not sufficient' features. This explains why LogicAl-PT achieves superior out-of-distribution generalization performance, delivering more consistent results across diverse data distributions compared to its competitors.
> >
> > [3] Selvaraju, Ramprasaath R., et al. "Grad-cam: Visual explanations from deep networks via gradient-based localization." Proceedings of the IEEE international conference on computer vision. 2017.
> >
> > &nbsp;
> >
> > >**Q2: Usefulness of the textual logic alignment.**
> >
> > **Answer:**
> >
> > **1) Qualitative Analysis:** Since the textual representations (corresponding to variable $\Phi_t$) are the class-wise mapping from the text labels, the sufficiency of variable $Y$ for variable $\Phi_t$ (i.e., $Y\Rightarrow \Phi_t$) is naturally guaranteed while the reverse $Y\Leftarrow \Phi_t$ is not ensured. In other words, textual representations ($\Phi_t$) must be necessary causes for variable $Y$, but they don't have to be sufficient causes for variable $Y$. Therefore, textual logic alignment is proposed to enhance the sufficiency of text representations ($\Phi_t$) for label $Y$. Accordingly, when cross-modal logic alignment (i.e., $\Phi_t \Leftrightarrow \Phi_v$) is achieved, combining textual logic alignment can mitigate the visual features that are not sufficient for variable $Y$.
> >
> > **2) Experimental Validation:** To investigate the actual role that textual logic alignment serves, we visualize the features which is utilized by the model tuned without textual logic alignment (w/o TLA), i.e., $\beta=0$. In particular, when we set $\beta=0$, $\alpha$ is tuned to its optimal value, i.e., the cross-modal logic alignment ($\Phi_t \Leftrightarrow \Phi_v$) is enhanced. **The visualization results are displayed in Figure 3 on page 10**. From the visualization results, we find that adding textual logic alignment mitigates the visual features which are not sufficient for predicting $Y$. **Therefore, the above qualitative analysis is validated by the visualization results.**
> >
> > Detailed visualization results and analysis are provided on line 483-513, page 10.
> >
> > &nbsp;
> >
> > >**Q3: How to extractive NSC features in textual and visual modality.**
> >
> > **Answer:** Sorry, we missed providing the explanation for the NSC feature in Figure 1. "NSC" represents "necessary and sufficient cause''. Specifically, the NSC features in textual and visual modalities are given by $f([Q,CLASS])$ and $h(g(X))$, respectively. The interventions in textual and visual modalities are given by $f([\bar{Q},CLASS])$ and $\bar{h}(g(X))$, respectively.
> >
> > At the training stage, "NSC" features are optimized by adjusting the learnable prompt $Q$ and filter $h$ using the proposed objective (10), as stated on line 290, page 6.
> >
> > At the inference stage, predictions are made using the cosine similarity between textual and visual "NSC" features.
> >
> > We have updated illustrations and descriptions of the overall framework in **Figure 1, on page 5**, in the revised paper.
> >
> > &nbsp;
> >
> > Thank you again for your valuable feedback. Further discussions are always welcome if you have any additional concerns or questions.

---

> > > ### Author Response · Authors · 2024-11-30
> > > **Looking forward to the reviewer's responses**
> > >
> > > Dear Reviewer,
> > >
> > > Thank you very much for taking the time to provide constructive and valuable comments on this work, which have significantly contributed to improving the quality of the paper.
> > >
> > > As the discussion period nears its conclusion, we would like to know if there are any additional clarifications or experiments we can provide. We look forward to your feedback and kindly invite you to update your score if your concerns have been adequately addressed.
> > >
> > > Thank you once again for your time!
> > >
> > > Best Regards,
> > >
> > > The Authors

---

> > > ### Author Response · Authors · 2024-12-02
> > > **Looking forward to the reviewer's responses**
> > >
> > > Dear Reviewer Rh7Y,
> > >
> > > As the Reviewer-Author discussion phase **concludes at midnight today (23:59, Dec 2nd, AoE)**, we kindly ask if we have adequately addressed your questions and concerns. Further discussions are welcome if you have any additional concerns or questions.
> > >
> > > Thank you once again for your time!
> > >
> > > Best Regards,
> > >
> > > The Authors

---

### Official Review · Reviewer_zmpK · 2024-10-30

**Soundness:** 3
**Presentation:** 3
**Contribution:** 2
**Rating:** 6
**Confidence:** 4

**Summary:**

This paper introduces the novel LogicAI-PT framework to mitigate learning of spurious correlations in prompt tuning of CLIPs. It models the PNS (probability of necessity and sufficiency) by introducing intervention $\bar{Q}$ and $\bar{\Phi}$. The author provide extensive introduction of the methodology and background and the experimental results are significant.

**Strengths:**

The proposed method is quite simple and effective regarding the significant improvement of multiple benchmarks. The idea of tackling spurious correlation problem from a sufficiency-necessity view is intuitive and is implemented based on thorough proof.

**Weaknesses:**

1. The proposed method seems to be universally applicable to many tasks rather than only prompting of VLM as classifier. Causal Representation Learning baselines adapting from other tasks can largely consolidate the motivation of this paper.
2. Ablation study of the proposed is not enough. What is the effect of different $\alpha, \beta$? #419 introduces the author chooses different value combination for different benchmarks. The robustness regarding different hyper-parameters can largely affect the applicability of this method.

**Questions:**

1. What kind of knowledge does the $\bar{Q}$ and $\bar{\Phi}$ learnt durning the training process? I am curious about more quantitative and qualitative results about these additionally introduced intervention modules.
2. Could this method transfer to other VLMs such as EVA-CLIP and single tower VLMs such as BEiT?

---

> ### Author Response · Authors · 2024-11-26
> **Response to Reviewer zmpK (Part 1)**
>
> Thank you very much for your valuable comments and feedback, which significantly contributes to improving the quality of this paper. Detailed responses to your concerns and questions are listed below.
>
> >**W1: Causal Representation Learning baselines adapting from other tasks can largely consolidate the motivation of this paper.**
>
> **Answer:** We adapt two representative causal representation learning methods from invariant learning: IRM [1] and IB-IRM [2]. They mitigate spurious correlations by ensuring the invariance of the conditional probability of the label $Y$ given the causal representation across varied training environments. The evaluation is conducted using ResNet-50 as the backbone model. The experimental results on four datasets are list as follows:
>
> | Dataset                   | Waterbird | | CelebA | | ImageNet | | PACS | |
> | :---                          | :---: | :---: | :---: | :---: | :---: | :---: | :---: | :---: |
> | Test Acc (%)           | Worst-case | Average | Worst-case | Average | Worst-case | Average | Worst-case | Average |
> | ERM                       | $54.7$ | $84.1$ | $26.7$ | $78.2$ | $80.5$ | $88.5$ | $80.0$ | $92.6$ |
> | IRM [1]                   | $64.7$ | $83.9$ | $67.1$ | $86.2$ | $87.9$ | $93.6$ | $80.7$ | $\mathbf{93.8}$ |
> | IB-IRM [2]              | $65.3$ | $84.3$ | $67.9$ | $85.8$ | $88.3$ | $93.9$ | $81.2$ | $93.4$ |
> | LogicAl-PT             | $\mathbf{67.5}$ | $86.2$ | $\mathbf{69.9}$ | $\mathbf{87.3}$ | $\mathbf{90.2}$ | $\mathbf{95.1}$ | $\mathbf{82.4}$ | $93.7$ |
>
> [1] Arjovsky, Martin, et al. "Invariant risk minimization." arXiv preprint arXiv:1907.02893 (2019).
>
> [2] Ahuja, Kartik, et al. "Invariance principle meets information bottleneck for out-of-distribution generalization." Advances in Neural Information Processing Systems 34 (2021): 3438-3450.
>
> **Analysis:** We can find that LogicAl-PT outperforms the typical causally invariant representation learning methods. The underlying reason stems from the advantage of 'sufficient and neccesary' causal representation over traditional causal representation, which also forms the motivation for proposing LogicAl-PT for prompt tuning of VLMs. Prevalent causal representation learning methods primarily aim to mitigate non-causal spurious correlations. In contrast, the concept of 'sufficiency and necessity' goes further by excluding not only non-causal spurious correlations but also causal relationships that are 'sufficient but not necessary' or 'necessary but not sufficient'. We provide specific examples to clarify these types of relationships and explain why only 'sufficient and necessary' relations remain stable across diverse data distributions in Figure 4 on page 14 in Appendix A.
>
> &nbsp;
>
> >**W2: Ablation study of the proposed is not enough. What is the effect of different $\alpha$, $\beta$?**
>
> **Answer:** We add experiments to evaluate the effects of two significant hyper-parameters in the proposed objective (i.e., $\alpha$ and $\beta$) on model performance. Since the results on other datasets present the similar tendency as on ImageNet-1K, we herein focus on ImageNet-1K, with ResNet-50 as backbone model. When evaluating the effect of $\alpha$, we fix $\beta=1.0$ . When evaluating the effect of $\alpha$, we fix $\alpha=20.0$. The experimental results are shown in the following two tables:
>
> | $\alpha$                 |  $0.0$ | $1.0$ | $10.0$ |  $20.0$ | $30.0$  | $50.0$  |
> | :---                         |    :---:   | :---:   | :---:      | :---:       | :---:       | :---:      |
> | worst-case (%) | $78.6$ | $80.9$ | $86.1$ | $90.2$ | $88.7$ | $79.5$ |
> | average (%)      | $87.2$ | $89.4$ | $93.5$ | $95.1$ | $94.0$ | $87.9$ |
>
> | $\beta$                  | $0.0$   | $0.10$ | $1.00$ | $10.0$ | $20.0$ | $30.0$ |
> | :---                         |    :---:   | :---:      |   :---:    |  :---:    | :---:      | :---:      |
> | worst-case (%) | $88.6$ | $89.7$ | $90.2$ | $89.3$ | $87.2$ | $85.5$ |
> | average (%)      | $94.3$ | $94.8$ | $95.1$ | $94.5$ | $93.2$ | $91.9$ |
>
> **Analysis:** When $\alpha=0.0$, model is tuned with only textual logic alignment; when $\beta=0.0$, models is tuned with only cross-modal logic alignment. We can find the performance of LogicAl-PT is more sensitive to the selection of $\alpha$ than the selection of $\beta$. To effectively mitigate spurious correlations in VLMs, careful tuning of $\alpha$ is essential. Regarding $\beta$, a small value is safer in practice, as a large $\beta$ may compromise the discriminative capability of the extracted features.

---

> > ### Author Response · Authors · 2024-11-26
> > **Response to Reviewer zmpK (Part 2)**
> >
> > >**Q1: What kind of knowledge does the $\mathbf{\bar{Q}}$ and $\mathbf{\bar{\Phi}}$ learnt durning the training process?**
> >
> > **Answer:** That's a very intriguing question. We have attempted to conduct visualization experiments to interpret the knowledge $\bar{Q}$ and $\bar{\Phi}$ learned during the training process. Unfortunately, we were unable to obtain understandable and meaningful visualization results. However, we believe that interpreting these two intervention modules is highly intriguing and warrants further investigation.
> >
> > >**Q2: Could this method transfer to other VLMs such as EVA-CLIP and single tower VLMs such as BEiT?**
> >
> > **Answer:** Yes, the proposed method could be adapted for other VLMs. However, determining how to implement this transfer and how to leverage PNS modeling to enhance the training of other VLMs remains a challenging task. We consider exploring the application of the proposed method to a broader range of VLMs as a future research direction.

---

> > > ### Comment · Reviewer_zmpK · 2024-11-27
> > >
> > > Thanks for the response. It solves my concerns and I will raise my score.

---

> > > > ### Author Response · Authors · 2024-11-27
> > > > **Thanks for the re-evaluation**
> > > >
> > > > Thank you very much for your efforts and response. We are pleased to have addressed your concerns, and truly appreciate your re-evaluation of our work and the decision to raise the score. Your valuable feedback has greatly facilitated the improvement of our work. Further discussions are welcome if you have any additional concerns or questions.

---

### Official Review · Reviewer_ZvzB · 2024-11-04

**Soundness:** 3
**Presentation:** 3
**Contribution:** 3
**Rating:** 6
**Confidence:** 2

**Summary:**

This paper presents Logical-pt, a framework for mitigating spurious correlations in vision-language models. It uses causally motivated logic alignment to align visual and textual features during prompt tuning. The method is backed by a tighter generalization error bound and empirically validated on several datasets, outperforming existing methods in out-of-distribution generalization.

**Strengths:**

* the results reported in this paper demonstrates good performance on multiple benchmarks
* this paper did extensive evaluations and experiments to validate the method's effectiveness

**Weaknesses:**

* The proposed method shows superior performance compared with other benchmarks. What is the computational efficiency compare to simpler methods?

**Questions:**

same as above

---

> ### Author Response · Authors · 2024-11-26
> **Response to Reviewer ZvzB**
>
> Thank you very much for your valuable feedback and suggestions, which significantly contributes to improving the quality of this paper. Detailed responses to your concerns and questions are listed below.
>
> >**W1: The proposed method shows superior performance compared with other benchmarks. What is the computational efficiency compare to simpler methods?**
>
> **Answer:** Thank you for your appreciation of our work. Regarding computational efficiency, we analyze and compare the computational overhead of our method with several existing methods. The results are presented in the following table.
>
> | Method      |  Params  | Params+%CLIP | FLOPS | FLOPS+%CoOp |
> | :---             | :---:          | :---:                      | :---:       | :---:                   |
> | CoOp         | 2048       | 0.004%               | 354.50G | -                        |
> | ERM           | 0.514M   |  1.05%                | 354.53G | 0.01%              |
> | CoOPood   | 1.026M   | 2.10%                | 354.56G  | 0.02%              |
> | LogicAl-PT | 1.028M   | 2.10%                 | 354.57G | 0.02%.             |
>
> **Analysis:** We can see the the overall parameters and Floating Point Operations (FLOPS) of our LogicAl-PT are only 2.1% and 0.02% higher than those of CLIP and CoOp, respectively. Compared with the improvement in out-of-distribution generalization performance, our method LogicAl-PT exhibits considerable computational efficiency in terms of the number of parameters and FLOPS.

---

> > ### Comment · Reviewer_ZvzB · 2024-11-29
> >
> > Thanks! This is helpful.

---

> > > ### Author Response · Authors · 2024-11-30
> > >
> > > Thank you very much for your responses. Further discussions are always welcome if you have any additional concerns or questions.

---

### Official Review · Reviewer_fk2Q · 2024-11-04

**Soundness:** 2
**Presentation:** 3
**Contribution:** 2
**Rating:** 5
**Confidence:** 3

**Summary:**

The paper presents a novel framework, LogicAl-PT, that addresses the challenge of cross-modal mitigation of spurious correlations in prompt tuning of vision-language models. The authors introduce a new concept, logic alignment, which integrates the mitigation of spurious correlations with cross-modal alignment of representations, and demonstrates its effectiveness through theoretical analysis and empirical results on various out-of-distribution datasets. LogicAI-PT earns competitive performance compared with traditional prompt-tuning methods for CLIP model.

**Strengths:**

Extensive theoretical analysis is provided to verify the proposed concept, PNS and PNS risk modeling.

**Weaknesses:**

- In contrast to the detailed theoretical analyses, the empirical verifications are fairly absent in this paper. Take the most recent competitor Coopood as an example, this paper presents much fewer empirical analyses, i.e. only 2 tables in the experiment section for verification. More ablation studies about the hyper-parameter chosen, visual results about the improvement on spurious correlations should be provided.

-Some more recent prompt tuning methods[a][b] should be discussed and compared with.

- Experiments on more architectures like ViT except for ResNet-50 should be done as well.

[a] Self-regulating Prompts: Foundational Model Adaptation without Forgetting, https://arxiv.org/abs/2307.06948)
[b] DePT: Decoupled Prompt Tuning, https://arxiv.org/abs/2309.07439

**Questions:**

Please see weakness.

---

> ### Author Response · Authors · 2024-11-26
> **Response to Reviewer fk2Q (Part 1)**
>
> Thank you very much for your valuable feedback and suggestions, which significantly contributes to improving the quality of this paper. Detailed responses to your concerns and questions are listed below.
>
> >**W1: Ablation studies about the hyper-parameter chosen should be provided..**
>
> **Answer:** We add experiments to evaluate the effects of two significant hyper-parameters in the proposed objective (i.e., $\alpha$ and $\beta$) on model performance. Since the results on other datasets present the similar tendency as on ImageNet-1K, we herein focus on ImageNet-1K, with ResNet-50 as backbone model. When evaluating the effect of $\alpha$, we fix $\beta=1.0$ . When evaluating the effect of $\alpha$, we fix $\alpha=20.0$. The experimental results are shown in the following two tables:
>
> | $\alpha$                 |  $0.0$ | $1.0$ | $10.0$ |  $20.0$ | $30.0$  | $50.0$  |
> | :---                         |    :---:   | :---:   | :---:      | :---:       | :---:       | :---:      |
> | worst-case (%) | $78.6$ | $80.9$ | $86.1$ | $90.2$ | $88.7$ | $79.5$ |
> | average (%)      | $87.2$ | $89.4$ | $93.5$ | $95.1$ | $94.0$ | $87.9$ |
>
> | $\beta$                  | $0.0$   | $0.10$ | $1.00$ | $10.0$ | $20.0$ | $30.0$ |
> | :---                         |    :---:   | :---:      |   :---:    |  :---:    | :---:      | :---:      |
> | worst-case (%) | $88.6$ | $89.7$ | $90.2$ | $89.3$ | $87.2$ | $85.5$ |
> | average (%)      | $94.3$ | $94.8$ | $95.1$ | $94.5$ | $93.2$ | $91.9$ |
>
> **Analysis:** When $\alpha=0.0$, model is tuned with only textual logic alignment; when $\beta=0.0$, models is tuned with only cross-modal logic alignment. We can find the performance of LogicAl-PT is more sensitive to the selection of $\alpha$ than the selection of $\beta$. To effectively mitigate spurious correlations in VLMs, careful tuning of $\alpha$ is essential. Regarding $\beta$, a small value is safer in practice, as a large $\beta$ may compromise the discriminative capability of the extracted features.
>
> &nbsp;
>
> >**W2: Visual results about the improvement on spurious correlations should be provided.**
>
> **Answer:** We add visualization experiment in the updated paper.
>
> **Setup:** For the purpose of verifying that the tuned models developed by our method LogicAl-PT exploit the necessary and sufficient features rather than spurious features, we sample some data instances to generate visual explanations for the selected model using Grad-CAM [1]. The commonly used Grad-CAM can produce a localization map which highlights the important regions in the input image that a deep learning model depends on for predicting the label.
>
> **Results: The detailed visualization results are displayed in Figure 2 (on page 9)**. The pivotal features employed by various prompt tuning methods are highlighted in red. **The visualization results reveal that the proposed LogicAl-PT demonstrates three notable advantages** over existing prompt-tuning methods: **1) LogicAl-PT can effectively eliminate the non-causal spurious features** that are associated with the label (i.e., 'background' in WaterBird dataset and 'baby' in ImageNet-1K dataset). **2) LogicAl-PT can mitigate the 'sufficient but not necessary' features** that demonstrate inconsistent presence across different data instances. For example, the shape of feet is a 'sufficient but not necessary' feature for classifying the picture of a bird as 'waterbird' or 'landbird' because its feet can retract or remain hidden when the bird is lying down or in flight. **3) LogicAl-PT can mitigate the 'necessary but not sufficient' features** which can impact the classification performance when the distribution of these 'necessary but not sufficient' features varies. For example, the wings of birds are 'necessary but not sufficient' features for distinguishing 'waterbird' from 'landbird'.
>
> **Analysis:** In summary, visualization results demonstrate the proposed LogicAl-PT can effectively exploit the 'sufficient and necessary' features and mitigate the unstable features, including non-causal spurious features, 'sufficient but not necessary' features and 'necessary but not sufficient' features. **This explains why LogicAl-PT achieves superior out-of-distribution generalization performance**, delivering more consistent results across diverse data distributions compared to its competitors.
>
> [1] Selvaraju, Ramprasaath R., et al. "Grad-cam: Visual explanations from deep networks via gradient-based localization." Proceedings of the IEEE international conference on computer vision. 2017.

---

> > ### Author Response · Authors · 2024-11-26
> > **Response to Reviewer fk2Q (Part 2)**
> >
> > >**W3: Experiments on more architectures like ViT except for ResNet-50 should be done as well.**
> >
> > **Answer:** We have added experiments on four commonly used datasets with ViT-B/32 as the backbone model. The experimental results are shown in the following table.
> > | Dataset                   | Waterbird | | CelebA | | ImageNet | | PACS | |
> > | :---                          | :---: | :---: | :---: | :---: | :---: | :---: | :---: | :---: |
> > | Test Acc (%)           | Worst-case | Average | Worst-case | Average | Worst-case | Average | Worst-case | Average |
> > |    CLIP                     | $41.4$ | $65.3$ | $69.7$ | $85.2$ | $51.4$ | $75.8$ | $81.7$ | $93.8$ |
> > |    CoOp                    | $43.5$ | $77.4$ | $26.2$ | $77.0$ | $87.1$ | $92.8$ | $82.4$ | $94.5$ |
> > |    ERM                     | $49.6$ | $78.3$ | $25.9$ | $76.8$ | $86.7$ | $93.3$ | $82.9$ | $94.1$ |
> > |  CoOPood               | $52.5$ | $79.2$ | $27.1$ | $76.5$ | $89.9$ | $94.6$ | $82.7$ | $94.4$ |
> > | PromptSRC             |  $50.8$ | $79.5$ | $69.3$ | $85.9$ | $87.8$ | $94.1$ | $83.4$ | $94.8$ |
> > | DePT+PromptSRC  | $51.7$ | $80.0$ | $70.2$ | $86.3$ | $87.4$ | $94.3$ | $83.5$ | $95.1$ |
> > |   LogicAl-PT            | $\mathbf{61.2}$ | $\mathbf{80.3}$ | $\mathbf{73.1}$ | $\mathbf{86.9}$ | $\mathbf{91.8}$ | $\mathbf{95.4}$ | $\mathbf{84.3}$ | $\mathbf{95.2}$ |
> >
> > **Analysis:** The results show that our method LogicAl-PT consistently outperforms the competitors on both worst-case and average test accuracy in four commonly used datasets. In particular, LogicAl-PT achieves around 9%, 3%, 2% and 1% higher worst-case accuracy than the second best algorithm on Waterbird, CelebA, ImageNet-1K and PACS when ViT-B/32 is used as backbone model.
> >
> > &nbsp;
> >
> > >**W4: Some more recent prompt tuning methods[a] (PromptSRC) [b] (DePT) should be discussed and compared with.**
> >
> > **Answer:** We have added the suggested recent prompt tuning methods: PromptSRC and DePT as baselines.
> >
> > **1)** When adopting ResNet-50 as backbone model, the corresponding results are listed as follows:
> > | Dataset                   | Waterbird | | CelebA | | ImageNet | | PACS | |
> > | :---                          | :---: | :---: | :---: | :---: | :---: | :---: | :---: | :---: |
> > | Test Acc (%)           | Worst-case | Average | Worst-case | Average | Worst-case | Average | Worst-case | Average |
> > | CoOPood               | $60.3$ | $\mathbf{86.3}$ | $31.6$ | $78.6$ | $85.8$ | $92.9$ | $81.5$ | $92.8$ |
> > | PromptSRC            | $57.2$ | $85.5$ | $68.2$ | $85.3$ | $81.6$ | $89.4$ | $81.7$ | $93.6$ |
> > | DePT+PromptSRC | $57.9$ | $86.0$ | $68.3$ | $85.7$ | $82.0$ | $90.1$ | $81.6$ | $\mathbf{93.9}$ |
> > | LogicAl-PT             | $\mathbf{67.5}$ | $86.2$ | $\mathbf{69.9}$ | $\mathbf{87.3}$ | $\mathbf{90.2}$ | $\mathbf{95.1}$ | $\mathbf{82.4}$ | $93.7$ |
> >
> > **2)** When adopting ViT-B/32 as backbone model, the corresponding results are listed as follows:
> > | Dataset                   | Waterbird | | CelebA | | ImageNet | | PACS | |
> > | :---                          | :---: | :---: | :---: | :---: | :---: | :---: | :---: | :---: |
> > | Test Acc (%)           | Worst-case | Average | Worst-case | Average | Worst-case | Average | Worst-case | Average |
> > | CoOPood               | $52.5$ | $79.2$ | $27.1$ | $76.5$ | $89.9$ | $94.6$ | $82.7$ | $94.4$ |
> > | PromptSRC           | $50.8$ | $79.5$ | $69.3$ | $85.9$ | $87.8$ | $94.1$ | $83.4$ | $94.8$ |
> > | DePT+PromptSRC| $51.7$ | $80.0$ | $70.2$ | $86.3$ | $87.4$ | $94.3$ | $83.5$ | $95.1$ |
> > | LogicAl-PT            | $\mathbf{61.2}$ | $\mathbf{80.3}$ | $\mathbf{73.1}$ | $\mathbf{86.9}$ | $\mathbf{91.8}$ | $\mathbf{95.4}$ | $\mathbf{84.3}$ | $\mathbf{95.2}$ |
> >
> > **Analysis:** The results demonstrate that the proposed LogicAl-PT consistently achieves the highest worst-case test accuracy while maintaining comparable average test accuracy to recent prompt tuning methods. In contrast to PromptSRC and DePT, which lack specific designs for mitigating spurious correlations, LogicAl-PT effectively leverages the 'sufficient and necessary' features and mitigates the unstable features, including non-causal spurious features, ‘sufficient but not necessary’ features and ‘necessary but not sufficient’ features. This explains why LogicAl-PT achieves superior out-of-distribution generalization performance, providing more consistent results across diverse data distributions compared to competing methods.

---

> > > ### Author Response · Authors · 2024-11-30
> > > **Looking forward to the reviewer's responses**
> > >
> > > Dear Reviewer
> > >
> > > Thank you very much for taking the time to provide constructive and valuable comments on this work, which have significantly contributed to improving the quality of the paper.
> > >
> > > As the discussion period nears its conclusion, we would like to know if there are any additional clarifications or experiments we can provide. We look forward to your feedback and kindly invite you to update your score if your concerns have been adequately addressed.
> > >
> > > Thank you once again for your time!
> > >
> > > Best Regards,
> > >
> > > The Authors

---

> > > > ### Author Response · Authors · 2024-12-02
> > > > **Looking forward to the reviewer's responses**
> > > >
> > > > Dear Reviewer fk2Q,
> > > >
> > > > As the Reviewer-Author discussion phase **concludes at midnight today (23:59, Dec 2nd, AoE)**, we kindly ask if we have adequately addressed your questions and concerns. Further discussions are welcome if you have any additional concerns or questions.
> > > >
> > > > Thank you once again for your time!
> > > >
> > > > Best Regards,
> > > >
> > > > The Authors

---

### Author Response · Authors · 2024-11-28
**Common Responses**

We sincerely thank all the reviewers for their efforts and valuable feedback, which have greatly contributed to improving the quality of our work. In particular, we appreciate the reviewers' recognition of our:

**(1) novel and intuitive methodology design** (by reviewer **Rh7Y**, **zmpK**);

**(2) extensive and thorough theoretical analysis** (by Reviewer **fk2Q**, **zmpK**, **Rh7Y**);

**(3) significant performance improvement** on multiple benchmarks (by Reviewer **ZvzB**, **zmpK**).

To address the reviewers' concerns and questions, we have provided detailed explanations for unclear content and conducted extensive experiments to further validate the superiority of our method. Specifically, **we summarize the updated content included in the revised version of the paper as follows:**

>**Clarifications:**

**(1)** Updated illustrations and descriptions of the overall framework in **Figure 1, on page 5**.

**(2)** Added a detailed illustration to analyze the necessity and superiority of logic alignment in VLMs in **Appendix A, on page 14**.

>**Evaluations:**

**(1) Visualization experiments and analysis** that demonstrate the proposed LogicAl-PT effectively mitigates cross-modal spurious correlations and enhances logical alignment between visual representation and text label, in **Section 5.3 (on page 8-9)**.

**(2)** Added two state-of-the-art prompt tuning methods as **additional baselines** (i.e., PromptSRC and DePT), in **Section 5.2 (on page 8)**.

**(3)** Experiments on **another backbone model** (i.e., ViT-B/32), in **Section 5.2 (on page 8)**.

**(4)** **Visual explanation** to illustrate the necessity of the proposed textual logic alignment, in **Section 5.4 (on page 9-10)**.

**(5)** **Ablation study** on the sensitivity of hyper-parameters, in **Section 5.4 (on page 10)**.

**(6)** Experimental comparison with **causal representation learning baselines** adapted from single-modal scenarios, **on line 1126-1151, page 21-22, in Appendix D**.

**(7)** Evaluation of computational overhead to verify the **computational efficiency** of our method, **on line 1107-1124, page 21, in Appendix D**.

---

### Comment · Area_Chair_tVNG · 2024-12-02
**Dec 2 - last day for reviewers' questions**

Dear Reviewer fk2Q and Rh7Y,

This is a kind reminder that December 2 is the last day for reviewers to ask questions to authors. As the paper received diverse ratings and your initial ratings are negative, could you check the authors' responses by today and see whether the responses addressed your concerns? Your constructive and timely communications are strong contributions to the reviewing process.

Thank you,

AC

---

### Meta-Review · Area_Chair_tVNG · 2024-12-23

**Metareview:**

This paper aims to tackle the issue of cross-modal spurious correlation for parameter-efficient prompt tuning of VLMs. This work pointed out that cross-modal mitigation of spurious correlations during prompt tuning of vision-language models remains an open question, and further proposed the logic of logic alignment and a practical framework to calculate the probability of necessity and sufficiency (PNS) between the textual label and textual representations.

This paper recevied diverse ratings, i.e., 6, 6, 5, 3. The AC has read reviewers' comments, authors' responses, and the revised version.  The idea of calculating the probability of necessity and sufficiency and analyzing the cross-modal spurious correlation for the aspect of causal inference is novel. The main reasons for reject are that (1) the paper has been revised significantly to include more experimental results and quantitive analysis compared to the original version, which indicates that the submission is not fully ready for publication, (2) although the performance comparisons with PromptSRC and DePT have been included in the revision, the performance gap between the proposed method, PromptSRC and DePT are marginal, and the reasons on why the gap is marginal is not discussed. (3) The presentation of the paper is not good, which is not easy to follow the idea and contributions. Therefore, the AC does not recommend the current submission as accept.

**Additional Comments On Reviewer Discussion:**

Reviewers raised the concern on (1) a lack of quantitative analysis on the contribution of PNS, and (2) the workload of the original submitted paper may be questionable.

---

### Decision · Program_Chairs · 2025-01-22

Reject